# Malaria abrogates O'nyong–nyong virus pathologies by restricting virus infection in nonimmune cells

Anthony Torres-Ruesta[1,2], Teck-Hui Teo[1], Yi-Hao Chan[1], Siti Naqiah Amrun[1], Nicholas Kim-Wah Yeo[1], Cheryl Yi-Pin Lee[1], Samantha Yee-Teng Nguee[1], Matthew Zirui Tay[1], Francois Nosten[3,4], Siew-Wai Fong[1], Fok-Moon Lum[1], Guillaume Carissimo[1], Laurent Renia[1,5,6,7], Lisa FP Ng[1,2,8,9]

**O'nyongnyong virus (ONNV) is a re-emerging alphavirus previously known to be transmitted by main malaria vectors, thus suggesting the possibility of coinfections with arboviruses in co-endemic areas. However, the pathological outcomes of such infections remain unknown. Using murine coinfection models, we demonstrated that a preexisting blood-stage *Plasmodium* infection suppresses ONNV-induced pathologies. We further showed that suppression of viremia and virus dissemination are dependent on *Plasmodium*-induced IFNγ and are associated with reduced infection of CD45⁻ cells at the site of virus inoculation. We further proved that treatment with IFNγ or plasma samples from *Plasmodium vivax*–infected patients containing IFNγ are able to restrict ONNV infection in human fibroblast, synoviocyte, skeletal muscle, and endothelial cell lines. Mechanistically, the role of IFNγ in restricting ONNV infection was confirmed in in vitro infection assays through the generation of an IFNγ receptor 1 α chain (IFNγR1)–deficient cell line.**

## Introduction

O'nyongnyong virus (ONNV) is an enveloped, positive-sense, single-stranded RNA virus that belongs to the *Alphavirus* genus of the Togaviridae family (1). It is closely related to other arthritogenic alphaviruses from the Semliki Forest antigen complex, such as chikungunya virus (CHIKV), Ross River virus (RRV), and Mayaro virus (MAYV) (2). ONNV pathology in humans is characterized by fever, maculopapular skin rash, myalgia, incapacitating polyarthralgia and to a lesser extent lymphadenopathy (1). The disease is generally self-limiting and resolves within some days, but symptoms such as polyarthralgia and myalgia may persist in a small proportion of the cases (3). ONNV was first isolated in 1959 in Gulu, Uganda (4), during an outbreak that lasted 3 yr (1959–1962) and involved more than two million cases (5). After this outbreak, ONNV caused two other major epidemics: one in south-central Uganda in 1996 (3, 6) and another in Liberia and Ivory Coast in 2003 involving thousands of cases (7). More recently, epidemiological surveys have reported high seroprevalence of ONNV in Coastal Kenya (8) and Uganda (9), suggesting an underestimated burden of ONNV infections in sub-Saharan Africa.

The re-emergence and expansion of alphavirus infections in the tropics during the last decade introduce a new risk of coinfections with other highly prevalent endemic mosquito-borne diseases such as malaria. Limited serological studies in malaria-endemic countries in Africa have reported evidence of coinfections between *Plasmodium* parasites and alphaviruses. Specifically, anti-CHIKV antibodies were detected in a Nigerian cohort of *Plasmodium falciparum*-infected patients (10). In another study in Tanzania (11), a considerable proportion of febrile malaria patients were seropositive for CHIKV, suggesting pre-exposure or active CHIKV infection. However, these reports should be interpreted cautiously because antibodies against CHIKV and ONNV are highly cross-reactive (9).

ONNV and *Plasmodium* parasites share common anopheline vectors such as *Anopheles gambiae* and *Anopheles funestus* (1), thus increasing the likelihood of co-transmission. However, reports on ONNV and *Plasmodium* coinfections in humans are lacking despite the increasing rates of ONNV transmission and the overwhelming presence of malaria in Sub-Saharan Africa. This, together with the highly inflammatory signature of both infections (12, 13, 14, 15, 16), the detrimental role of T cell–mediated immunity in the pathologies induced by both infections (17, 18, 19, 20, 21), and the immunosuppressive nature of malaria (22, 23, 24, 25), strongly suggest that immune modulation could happen upon coinfection.

[1]A*STAR Infectious Diseases Labs (A*STAR ID Labs), Agency for Science, Technology and Research (A*STAR), Singapore, Singapore   [2]Department of Biochemistry, Yong Loo Lin School of Medicine, National University of Singapore, Singapore, Singapore   [3]Shoklo Malaria Research Unit, Mahidol-Oxford Tropical Medicine Research Unit (MORU), Faculty of Tropical Medicine, Mahidol University, Mae Sot, Thailand   [4]Nuffield Department of Medicine, Centre for Tropical Medicine and Global Health, University of Oxford, Oxford, UK   [5]Singapore Immunology Network, Agency for Science, Technology and Research (A*STAR), Singapore, Singapore   [6]Lee Kong Chian School of Medicine, Nanyang Technological University, Singapore, Singapore   [7]School of Biological Sciences, Nanyang Technological University, Singapore, Singapore   [8]National Institute of Health Research, Health Protection Research Unit in Emerging and Zoonotic Infections, University of Liverpool, Liverpool, UK   [9]Institute of Infection, Veterinary and Ecological Sciences, University of Liverpool, Liverpool, UK

Correspondence: lisa_ng@idlabs.a-star.edu.sg; renia_laurent@idlabs.a-star.edu.sg

In this study, we describe the interactions between ONNV and rodent *Plasmodium* parasites in a mammalian host. Using mouse models of coinfections, we demonstrated that preexisting murine malaria restricts ONNV-associated pathologies and this protective effect is driven mainly by *Plasmodium*-induced IFNγ by limiting ONNV infection in the CD45-cell compartment. In vitro experiments using human cell lines and plasma from *Plasmodium vivax*–infected patients confirmed the antiviral role of IFNγ in restricting ONNV infection. Our findings have potential implications in arbovirus and malaria control programs in endemic regions where *Plasmodium* parasites and arboviruses co-circulate.

## Results

### Murine malaria suppresses ONNV-induced joint swelling and viremia

An immunocompetent mouse model was previously established to recapitulate ONNV-induced joint pathologies (inflammation, edema, muscle necrosis, synovitis, and tenosynovitis) and acute viremia (26). Using this model, we first assessed whether a preexisting acute blood-stage *Plasmodium* infection could alter the development of ONNV pathologies. To do so, 3-wk-old C57BL/6J mice were inoculated with 1E6-infected red blood cells (iRBCs) from either *Plasmodium berghei* ANKA clone 231cl1 (PbA), which induces lethal neuropathology known as experimental cerebral malaria (ECM), or the nonlethal self-resolving strain *Plasmodium yoelii* 17XNL clone 1.1 (Py17x). When patent parasitemia was detected at 4 d post iRBC injection, 1E6 ONNV PFU were injected subcutaneously in the right paw. Viremia and joint swelling were measured for 12 and 14 days postinfection, respectively (dpi) (Fig 1A). The patent blood-stage Py17x infection protected coinfected animals from the development of ONNV-induced joint swelling (Figs 1B and S1A) and significantly reduced viremia levels (Fig 1C). Similarly, a preexisting blood-stage PbA infection was able to abolish virus-induced footpad swelling (Fig 1B). Of note, viremia in animals preinfected with PbA was undetectable during the entire follow-up suggesting a stronger protective effect by PbA than by Py17x against ONNV (Fig 1C).

To assess whether the suppression of viremia and ONNV-induced joint swelling by murine malaria was dependent on the timing of *Plasmodium* inoculation, two additional coinfection conditions were explored: concurrent coinfection and postviral coinfection. In concurrent coinfection, mice were coinfected with 1E6 ONNV PFU and 1E6 PbA or Py17x iRBC at the same time (Fig 1D). Concurrent coinfection did not affect the development of viremia in coinfected animals (Fig 1F) but had strain-specific effects on inflammation. Mice coinfected with Py17x did not display significant reduction in joint swelling, whereas mice coinfected with PbA displayed a significant reduction in joint swelling from 5 dpi onwards with a major suppression at the peak of swelling at 6 dpi and an earlier resolution of the pathology at 10 dpi compared with ONNV-infected controls (Figs 1E and S1B). Finally, we assessed postviral *Plasmodium* infection, where mice were infected with 1E6 PbA or Py17x iRBC 4 d after 1E6 ONNV PFU injection (Fig 1G). In this

setting, no effect on ONNV-induced joint swelling (Figs 1H and S1C) or viremia (Fig 1I) was observed, indicating that neither PbA nor Py17x infection could alter the course of a preexisting ONNV infection.

The effects of ONNV infection on the dynamics of malarial growth and survival were also assessed. The inoculation of ONNV 4 d post Py17x or PbA infection did not alter parasitemia levels (Fig S2A and B) or PbA-induced mortality (Fig S2B). Concurrent coinfection with nonlethal Py17x and ONNV resulted in increased parasitemia levels (Fig S2C). Py17x parasitemia resolution was delayed as coinfected animals took 26–28 d to clear blood-stage parasites compared with 20–22 d in controls. On the other hand, simultaneous inoculation with PbA and ONNV did not affect the development of parasitemia or ECM mortality in coinfected mice (Fig S2D). Finally, the infection with *Plasmodium* parasites 4 d post ONNV inoculation resulted in aggravated Py17x and PbA parasitemia (Fig S2E and F) but did not impact ECM mortality (Fig S2F).

### Early stages of ONNV replication in footpad tissues are impaired in animals preinfected by *Plasmodium* parasites

As shown in Fig 1C, a preexisting *Plasmodium* infection, either by PbA or Py17x, significantly reduced viremia in coinfected animals. To assess any possible differences in the kinetics of virus replication at the site of inoculation (right hind limb footpad), viral dissemination in vivo was assessed using a luciferase-tagged ONNV clone that mimics wild type ONNV infection in mice. Bioluminescence measurements from infected footpads correlate with viral burden in these tissues (26). ONNV-infected and coinfected animals were monitored during the first 24 hours postvirus inoculation (hpi) given that ONNV dissemination peaks at 12 hpi (31). Significant differences in whole body and footpad radiance were detected as early as 1 hpi in mice preinfected with PbA or Py17x and were maintained at 3, 6, 12, and 24 hpi (Fig 2A–C and Video 1).

The absence of ONNV bioluminescence signals into adjacent tissues such as the tail (Fig 2C) in the coinfected mice prompted us to further assess ONNV dissemination into other major mouse tissues. Appendages (hind limb footpads and tail), internal organs (spleen, liver, and pLN) and muscle tissue (gastrocnemius and quadriceps) of ONNV-infected and coinfected animals were harvested at 1 dpi and viral RNA was quantified. Lower viral loads were observed in most of the tissues assessed, with major differences occurring in organs distant from the site of inoculation such as liver and spleen where viral burden in coinfected mice were on average ≈10,000-fold and ≈1,000-fold lower, respectively (Fig 2D).

### Preexisting *Plasmodium* infection renders CD45$^+$ and CD45$^−$ footpad cells less susceptible to ONNV infection

Our data strongly suggested that a preexisting blood-stage *Plasmodium* infection renders footpad cells less susceptible to ONNV. To identify the cellular subsets involved in the suppression of ONNV infection, we defined the differences in ONNV infection of non-immune (CD45$^−$) and immune (CD45$^+$) cells in footpads of coinfected mice at 12 hpi (peak of footpad viral load in Fig 2B). For this purpose, a ZsGreen-tagged ONNV infectious clone detectable by flow cytometry under the FITC channel was used.

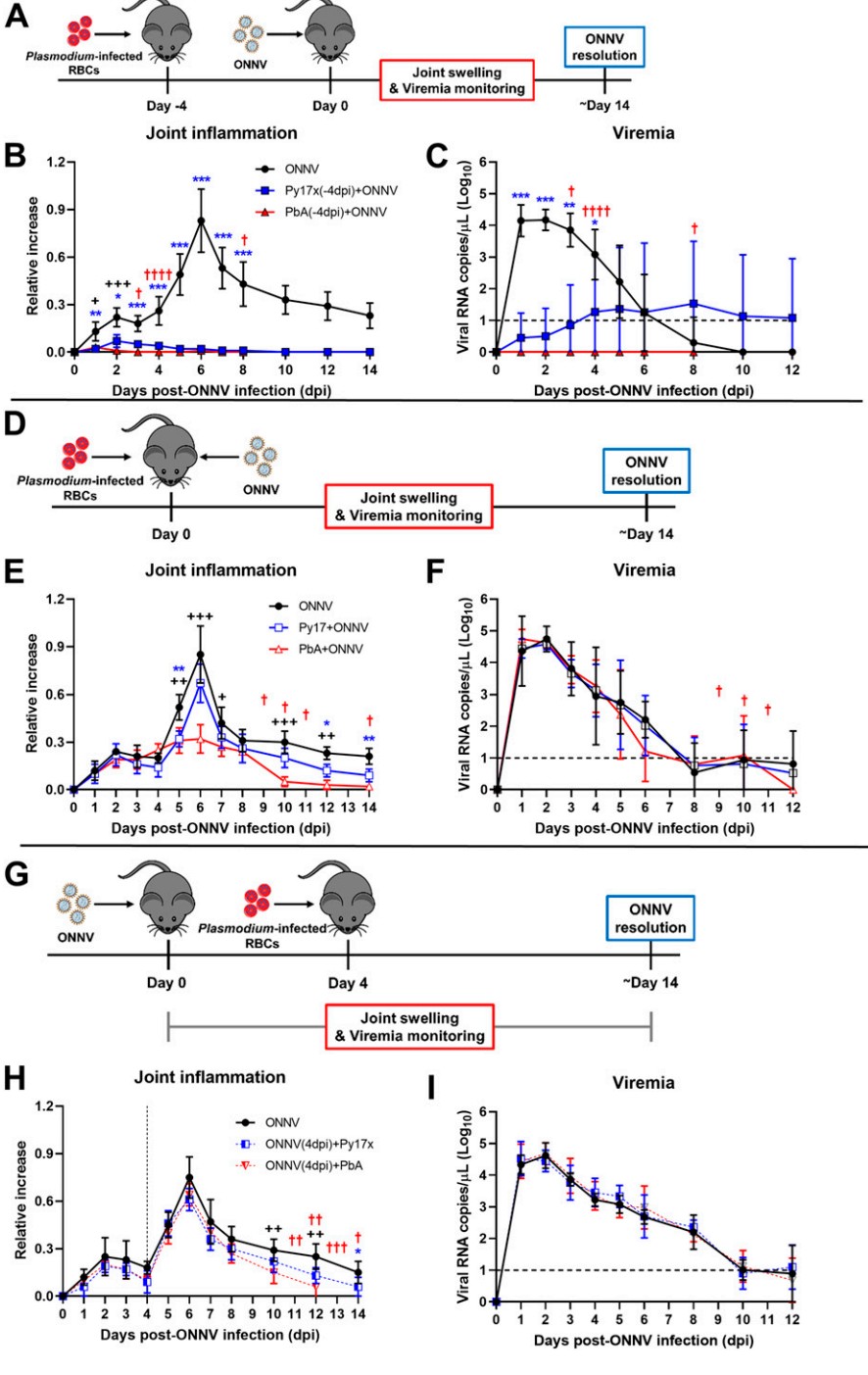

**Figure 1. Preexisting murine malaria protects mice from ONNV-induced pathologies.**
**(A)** Previral *Plasmodium* infection: mice were infected with PbA or Py17x 4 d prior ONNV inoculation according to the schematic in (A). **(B, C)** Joint swelling and (C) viremia measurements in ONNV, Py17x(-4dpi)+ONNV, and PbA(-4dpi)+ONNV groups. **(D)** Concurrent coinfection: animals were simultaneously infected with ONNV and PbA or Py17x on the same day according to the schematic in (D). **(E, F)** Joint swelling and (F) viremia measurements in ONNV, Py17x+ONNV, and PbA+ONNV groups. **(G)** Postviral *Plasmodium* infection: mice were infected with ONNV 4 d prior PbA or Py17x inoculation according to the schematic in (G). **(H, I)** Joint swelling and (I) viremia measurements in ONNV, ONNV(-4dpi)+Py17x, and ONNV(-4dpi)+PbA groups. Data are presented as mean ± SD of at least five animals per experimental group and are representative of two independent experiments. Differences between ONNV controls and coinfected mice with PbA (++$P$ < 0.01, +++$P$ < 0.01) or Py17x (*$P$ < 0.05, **$P$ < 0.01, ***$P$ < 0.001) were calculated using two-tailed Kruskal–Wallis and post hoc Dunn's tests. When PbA-infected animals succumbed to ECM, differences between ONNV singly infected controls and coinfected mice with Py17x were computed using two-tailed Mann–Whitney U test instead. "†" represents one mouse that succumbed of PbA-induced ECM on the respective day. Horizontal dashed lines in (C), (F) and (I) represent the qRT-PCR detection limit. Vertical dashed line in (H) represents the day on which *Plasmodium* parasites were inoculated.

To understand the individual contribution of CD45[+] and CD45[−] compartments to the total ONNV-infected cells at 12 hpi, the infectivity profile of footpads was analyzed in ONNV-infected mice. It was observed that ~80% of the ONNV-infected footpad cells at 12 hpi were part of the CD45[−] nonimmune compartment (Fig S3A). When the analysis was extended to median fluorescence intensity values, CD45[−] cells appeared to harbour on average a higher number of ONNV-ZsGreen copies than CD45[+] cells (Fig S3B). These

observations highlight a major role of CD45[−] cells as an early target for ONNV replication.

The impact of coinfections on the infectivity rates of CD45[+] and CD45[−] cells was then assessed. Animals preinfected with either PbA or Py17x displayed a marked reduction in the percentage and total counts of CD45[+]ZsGreen[+] and CD45[−]ZsGreen[+] cells compared with ONNV controls (Fig S3C and D). Given the importance of nonimmune cells as alphavirus targets during the early stages of infection

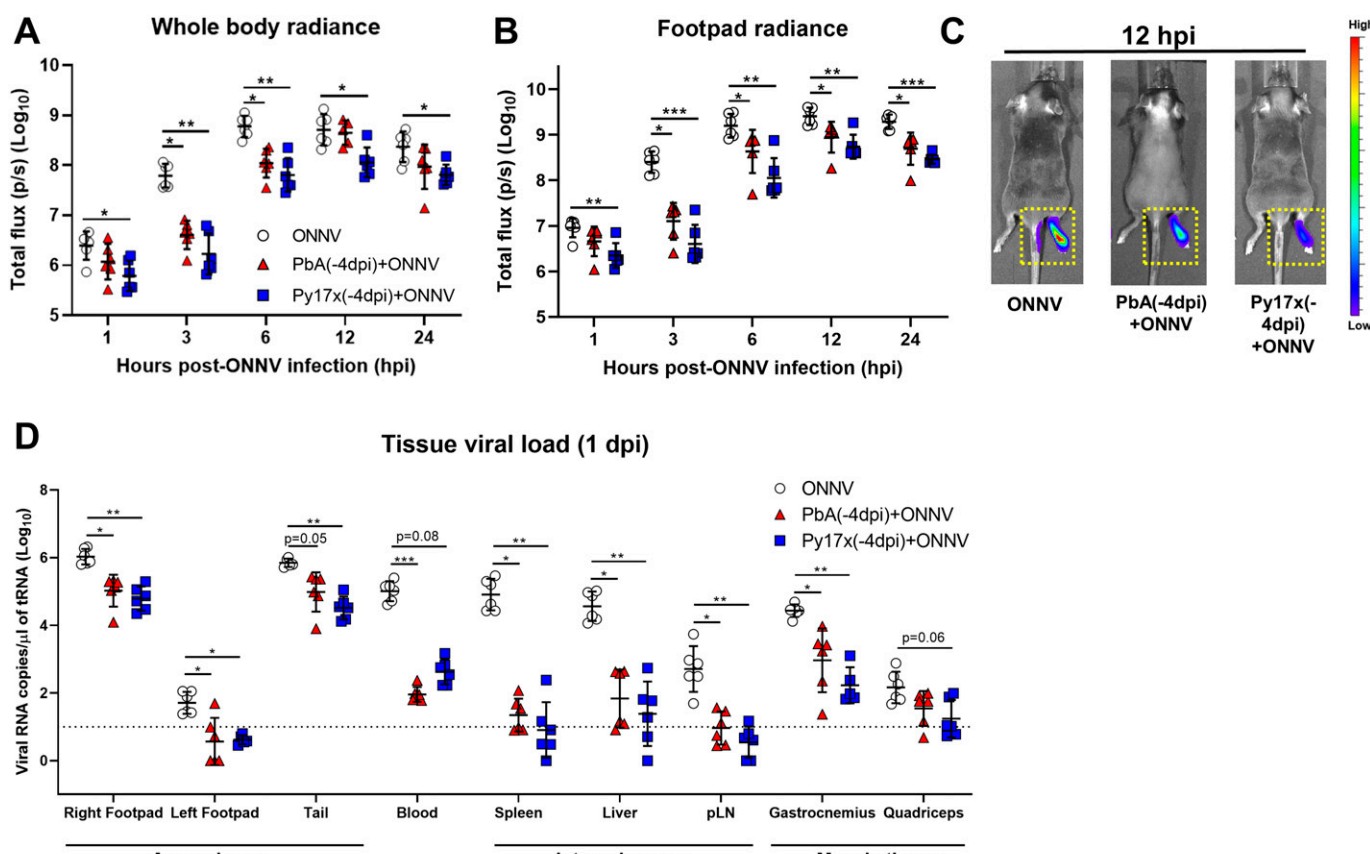

**Figure 2. Early stages of ONNV replication and dissemination are supressed by preexisting *Plasmodium* infection.**
Mice were infected with Py17x or PbA and 4 d postinfection were inoculated with a firefly luciferase-tagged ONNV clone in the right hind limb footpad. Before data acquisition, mice were injected with 100 μl of D-Luciferin (5 mg/ml) subcutaneously. **(A, B)** Whole body radiance and (B) footpad radiance of ONNV, Py17x(-4dpi)+ONNV, and PbA(-4dpi)+ONNV groups at 1, 3, 6, 12, and 24 hpi. **(C)** Representative pseudo-coloured images of bioluminescence readings depicting reduction of tissue viral load and restriction of viral dissemination in coinfected animals at 12 hpi (yellow-doted boxes). **(D)** Tissue viral load in mouse appendages, internal organs and muscle detected by qRT-PCR at 24 hpi. Data are presented as mean ± SD of at least five animals per experimental group. Differences between ONNV controls and coinfected mice with PbA or Py17x were calculated using two-tailed Kruskal–Wallis and post hoc Dunn's tests (*$P < 0.05$ **$P < 0.01$, ***$P < 0.001$).

(27, 28, 29), the differences in infectivity within the CD45⁻ compartment were further characterized using Uniform Manifold Approximation and Projection (UMAP) on the flow cytometry data. Manually gated CD45⁻ populations based on the expression of signature surface markers (30, 31, 32, 33, 34) were overlapped in the UMAP plot and allowed the identification of four major cell lineages: endothelial cells (CD45⁻CD29⁺CD31⁺), myoblasts (CD45⁻CD31⁻Sca-1⁺), fibroblasts (CD45⁻CD9⁺CD29⁺CD31⁻Sca-1⁻), and mesenchymal stem cells (CD45⁻CD9⁻CD29⁻CD31⁻Sca-1⁻) (Fig 3A). UMAP plots displaying the distribution of ZsGreen+ events in ONNV-infected and coinfected mice are also shown in Fig 3A.

As observed from the UMAP plots, ONNV infection was globally suppressed in endothelial cells, myoblasts, and fibroblasts from coinfected animals at 12 hpi (Fig 3A). A similar trend was observed when ZsGreen⁺ cells were quantified revealing a reduced number of infected endothelial cells, myoblasts, and fibroblasts (Fig 3B). To explore whether the reduced infectivity in CD45⁻ cells was a transient or maintained effect beyond 12 hpi, footpads from coinfected mice with Py17x were harvested at 48 and 72 hpi and analyzed by flow cytometry. Coinfected mice with PbA were not

included in these sets of experiments because of the high mortality rate induced by ECM. Interestingly, suppression of ONNV infection in coinfected animals was observed across all the different time points assessed, suggesting a sustained suppression of viral load and not a delay in viral load appearance (Fig 3C).

### Murine malaria up-regulates proinflammatory immune mediators in footpad tissues

Acute *Plasmodium* infections are known to induce strong proinflammatory responses characterized by the production and release of cytokines/chemokines and other mediators into the bloodstream to control parasite burden (15, 16). Thus, we hypothesized that some of these soluble factors could be responsible for the decreased susceptibility to ONNV infection observed in footpad tissues.

36 different immune mediators were assessed in sera (Fig 4A) and footpad tissue lysates (Fig 4B) of mice singly infected with lethal PbA or self-resolving Py17X at 4 d postparasite infection (time of ONNV inoculation in the coinfection model). Interestingly, results

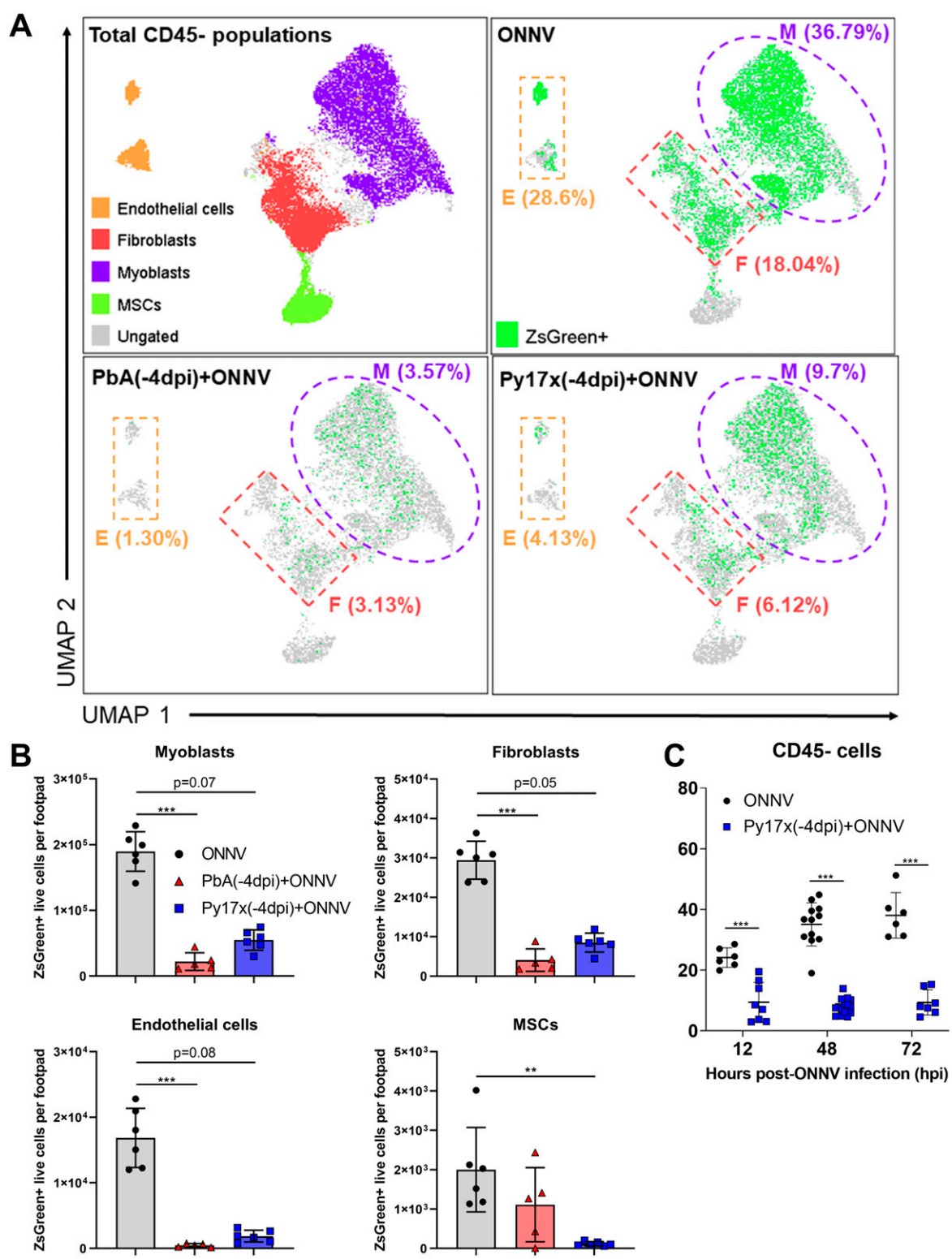

**Figure 3. Prior *Plasmodium* infection restricts ONNV replication in CD45⁻ footpad cells.**
**(A)** UMAP analysis of 105,000 live CD45⁻ footpad cells from naive, ONNV, PbA(-4dpi)+ONNV, and Py17x(-4dpi)+ONNV groups at 12 hpi. The UMAP plot was generated by concatenation of samples containing 5,000 randomly selected live CD45⁻ cells from each sample. ONNV, PbA(-4dpi)+ONNV, and Py17x(-4dpi)+ONNV UMAP plots show the global distribution of ZsGreen+ events (ONNV-infected cells). Colored dashed boxes highlight myoblasts (M), fibroblasts (F), and endothelial cells (E) and median ONNV infection rates per population. **(B)** Total counts of ONNV-ZsGreen+ cells in endothelial cells, myoblasts, fibroblasts and mesenchymal stem cells (MSCs) in ONNV, PbA(-4dpi)+ONNV, and Py17x(-4dpi)+ONNV at 12 hpi. **(C)** Percentages of infected CD45⁻ cells in ONNV and Py17x(-4DPI)+ONNV groups at 12, 48, and 72 hpi. Data are

showed that PbA- and Py17x-infected mice generated distinct pro-inflammatory cytokine profiles. Using principal component analysis on these cytokine profiles, a distinct separation was revealed between samples from mock (green), PbA-infected (red), and Py17x-infected mice (green) (Fig 4C).

Levels of pro-inflammatory IFNγ were increased by ~292- and ~28-folds in sera from PbA and Py17x-infected mice, respectively (Fig 4D), in line with previous reports (35, 36, 37). Up-regulation of chemokines such as CXCL10 (IFNγ-induced protein 10: IP-10), CXCL1, CCL2, CCL3, CCL5, and CCL7 was also observed in the sera (Fig 4D). Locally at the footpad, changes in immune mediator profiles were characterized by increased chemokine production, particularly IFNγ-induced CXCL10, which was found to be elevated by ~20- and ~11-folds in tissue lysates of PbA and Py17x-infected animals, respectively (Fig 4D). Other up-regulated chemokines shared between PbA and Py17x-infected mice were CCL2, CCL3, CCL4, CCL5, CCL7, and CCL11. In addition, IL-18 (IFNγ inducing factor: IGIF) and IFNγ were found to be slightly elevated in footpad tissues upon PbA and Py17x infection (Fig 4D).

Differentially regulated cytokines/chemokines in the footpads were subjected to STRING analysis (performed with a high confidence threshold of 0.9), revealing key interactions between eight of these immune mediators (Fig S4). The predicted interactions are linked to cellular responses to IFNγ, host-negative regulation of viral transcription and leukocyte recruitment and activation. Collectively, these data suggest that *Plasmodium* infection (either by PbA or Py17x) triggers a systemic immune response characterized by the up-regulation of pro-inflammatory immune mediators not only in the blood but also locally in footpad tissues. Importantly, the increased levels of immune mediators linked to IFNγ signaling, a known antiviral cytokine (38), in footpad lysates of *Plasmodium*-infected mice suggested a possible involvement of this cytokine in the restriction of ONNV infection.

### *Plasmodium*-induced IFNγ mediates the suppression of ONNV replication and dissemination in coinfected animals

To assess the potential role of IFNγ in suppressing ONNV pathologies, IFNγ-deficient animals were infected with nonlethal Py17x. 4 d postinfection, luciferase-tagged ONNV was inoculated subcutaneously in the right footpad. Lack of IFNγ in coinfected animals abolished the antiviral effects exerted by Py17x infection in mice. Bioluminescence readings at 3, 6, 12, and 24 hpi were comparable with those from IFNγ-deficient animals singly infected with ONNV (Fig 5A and Video 2). Similarly, ONNV viremia in coinfected IFNγ-deficient mice was also restored (Fig 5B) despite a suppression of joint swelling at 6 dpi (Fig S5). This suggests that coinfection might modulate other immune responses leading to joint swelling suppression in the absence of IFNγ.

Because nonimmune cells express IFNγ receptors (38) and can respond to IFNγ signaling, the infection profile of these subsets was assessed in coinfected mice deficient in IFNγ. Likewise, ONNV infectivity profiles of CD45⁻ cells were restored in myoblasts, endothelial

cells, fibroblasts, and mesenchymal stem cells (Fig 5C and D). To further validate these findings, in vivo IFNγ neutralization was performed using anti-mouse IFNγ antibodies. Coinfected animals treated with anti-mouse IFNγ displayed comparable ONNV tissue viral loads at 3, 6, 12, and 24 hpi (Fig 5E and Video 3) than isotype-control treated mice, suggesting a major role of IFNγ in the antiviral effects exerted by preexisting *Plasmodium* infections.

Blood-stage *Plasmodium* infections in humans and mice are known to induce the production of type I IFN responses (39, 40, 41). Type I IFN is a major regulator of susceptibility to alphavirus infection (13, 42, 43) and evidence has suggested crosstalk mechanisms with IFNγ signaling (44, 45, 46). To evaluate any possible contribution of type I IFN responses in the reduced susceptibility to ONNV upon coinfection, the effect of preexisting murine malaria on ONNV replication was assessed in IFNaR⁻/⁻ mice (deficient of IFN-α/β receptor). Viremia measurements at 12, 24, and 48 hpi in coinfected IFNaR⁻/⁻ mice (Fig S6) revealed that murine malaria was still able to restrict ONNV infection, ruling out the involvement of type I IFN responses in the antiviral effects exerted by *Plasmodium*-induced IFNγ.

### IFNγ inhibits ONNV infection in human fibroblast, synoviocyte, skeletal muscle, and endothelial cell lines

The antiviral role of IFNγ in nonimmune cell lineages upon ONNV infection remains poorly defined. To extrapolate our findings in the human context, the antiviral effect of IFNγ was assessed in human cell lines representing skin fibroblasts (BJ), synoviocytes (SW982), endothelial cells (HPMECs), and skeletal muscle cells (RD) in an in vitro infection system. Interestingly, each cell type displayed different susceptibility to ONNV infection. Particularly, skin fibroblast and synoviocyte cell lines are highly susceptible to ONNV, whereas endothelial cells and skeletal muscle cells are poorly infected. Nevertheless, regardless of cell type, IFNγ pretreatment successfully reduced ONNV infection in BJ, SW982, HPMEC, and RD cells at 24, 48, and 72 hpi in a dose-dependent manner (Fig 6A).

Finally, we characterized the ability of plasma samples from acute *P. vivax*–infected patients to render human cell lines less susceptible to ONNV infection. Plasma IFNγ levels were quantified, and samples were categorized and pooled into low (n = 13, median IFNγ concentration = 69.53 pg/ml) and high IFNγ producers (n = 14, median IFNγ concentration = 293.675 pg/ml). 10 plasma samples from healthy individuals were pooled and included in the experiments as controls (IFNγ concentration under the quantification limit) (Fig 6B). Incubation of skin fibroblasts (BJ) with plasma from either low or high IFNγ producers reduced ONNV infection compared with those treated with healthy control plasma (Fig 6C). To prove that IFNγ present in the plasma samples from acute *P. vivax*–infected patients was responsible for the antiviral effects exerted in vitro, we generated a HEK293T cell line with impaired IFNγ signaling (Fig S7A and B) by knocking down the expression of the IFNγ receptor 1 α chain (IFNγR1). Upon treatment with plasma from malaria patients, HEK293T cells with intact IFNγR1 expression

presented as mean ± SD of at least five animals per experimental group. Differences between ONNV controls and coinfected mice with PbA or Py17x were calculated using two-tailed Kruskal–Wallis and post hoc Dunn's tests (**P < 0.01, ***P < 0.001).

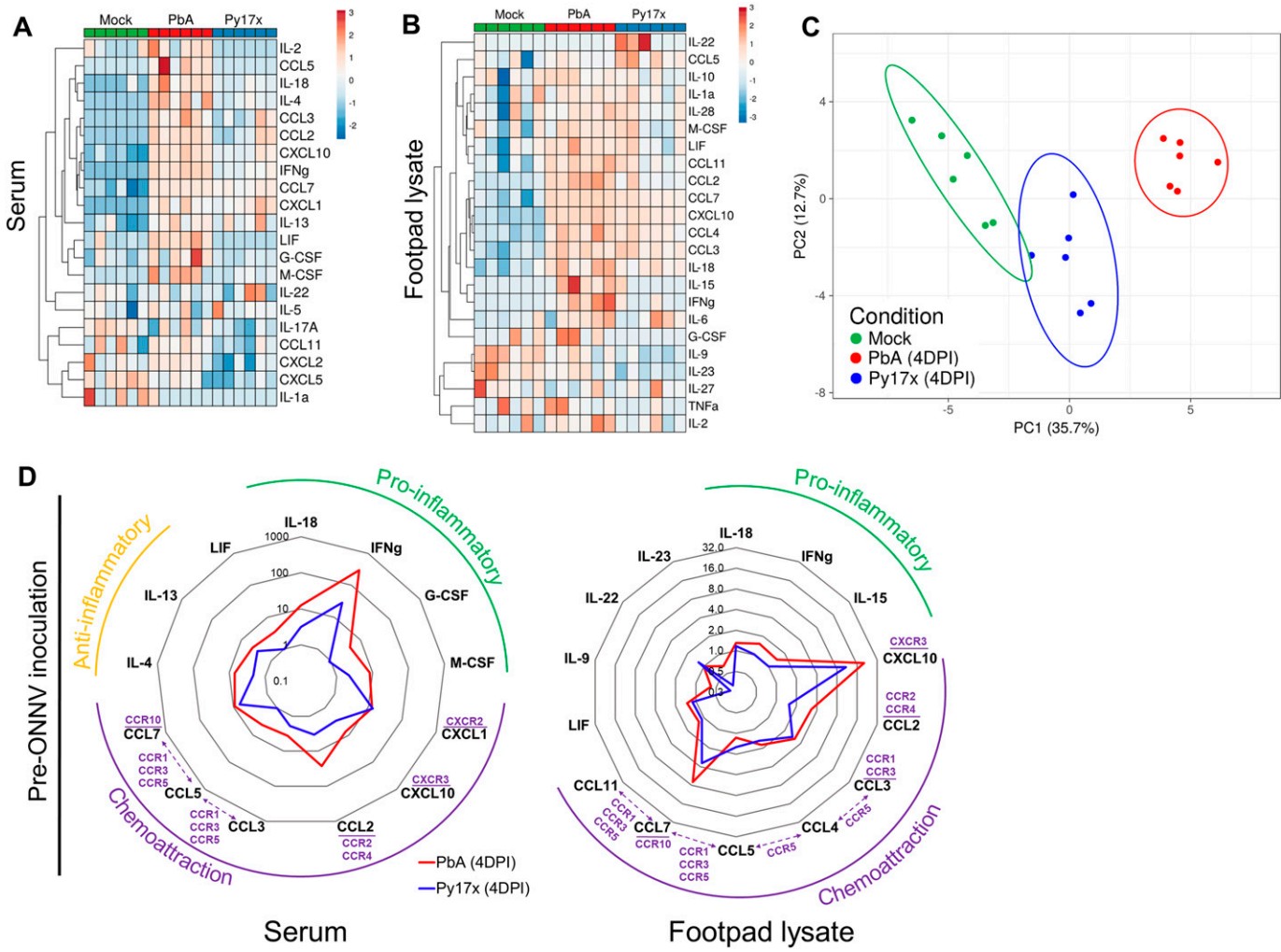

**Figure 4. Footpads of *Plasmodium*-infected mice display a pro-inflammatory milieu.**
**(A, B)** Heat map plots showing the detected cytokines/chemokines in (A) serum and (B) footpad lysates of mock-infected (green), PbA-infected (red), and Py17x-infected (blue) mice at 4 dpi. Analyte concentrations (pg/ml + 1) were logarithmically transformed (Log$_{10}$) and Z-scores were calculated for representation purposes. Principal component analysis (PCA) and heat map plots were constructed using ClustVis. **(C)** PCA using differentially expressed analytes in footpad lysates and sera of mock, PbA-infected (4 dpi), and Py17x-infected (4 dpi) groups. PCA plot shows that PC1 (responsible for 35.7% of the variation) and PC2 (responsible for 12.7% of the variation) segregate the populations in three clusters: mock (green), PbA-infected (red), and Py17x-infected (blue). Colored ellipses were calculated with 95% confidence levels. **(D)** Radar plots showing median fold changes of differentially expressed cytokines/chemokines in serum and footpad lysates of PbA-infected (4 dpi) and Py17x-infected (4 dpi) groups relative to mock animals. Each cytokine/chemokine is grouped according to its immunological function (green: pro-inflammatory, yellow: anti-inflammatory) or homing receptors (purple) as indicated. Shared chemokine receptors are shown in dashed lines. Data correspond six animals per experimental group. Differences between naïve, PbA, or Py17x-infected mice calculated using two-tailed Kruskal–Wallis and post hoc Dunn's tests.

displayed lower ONNV infection compared with untreated controls. This antiviral effect was lost in cells with impaired IFNγR1 expression (ΔIFNγR1) (Fig 6D), in agreement with an IFNγ-specific effect.

## Discussion

O'nyongnyong virus and *Plasmodium* parasites share common anopheline vectors and co-circulate in sub-Saharan Africa with risk of human coinfection. This is the first study investigating the pathological outcomes of coinfection by *Plasmodium* parasites and alphavirus ONNV in a mammalian host. Here, we showed that a preexisting murine *Plasmodium* infection is able to suppress the

development of ONNV pathologies by restricting viral infection at the site of inoculation and dissemination to distant organs. We demonstrated that *Plasmodium*-induced IFNγ is the main cytokine driving the antiviral effects observed.

IFNγ is a pleiotropic cytokine known for its ability to regulate immune responses by promoting macrophage activation, enhancing antigen presentation, modulating helper T cell development and mediating viral and bacterial immunity, among others (47). In ONNV-infected animals, it was observed that nearly 80% of the total virus-infected cells at 12 hpi (Fig S3A) belonged to the CD45⁻ compartment corroborating previous observations from other closely related alphaviruses, whereby non-immune cells support the early stages of viral replication (13, 27, 28, 29, 48). In contrast, animals harboring a *Plasmodium* infection displayed

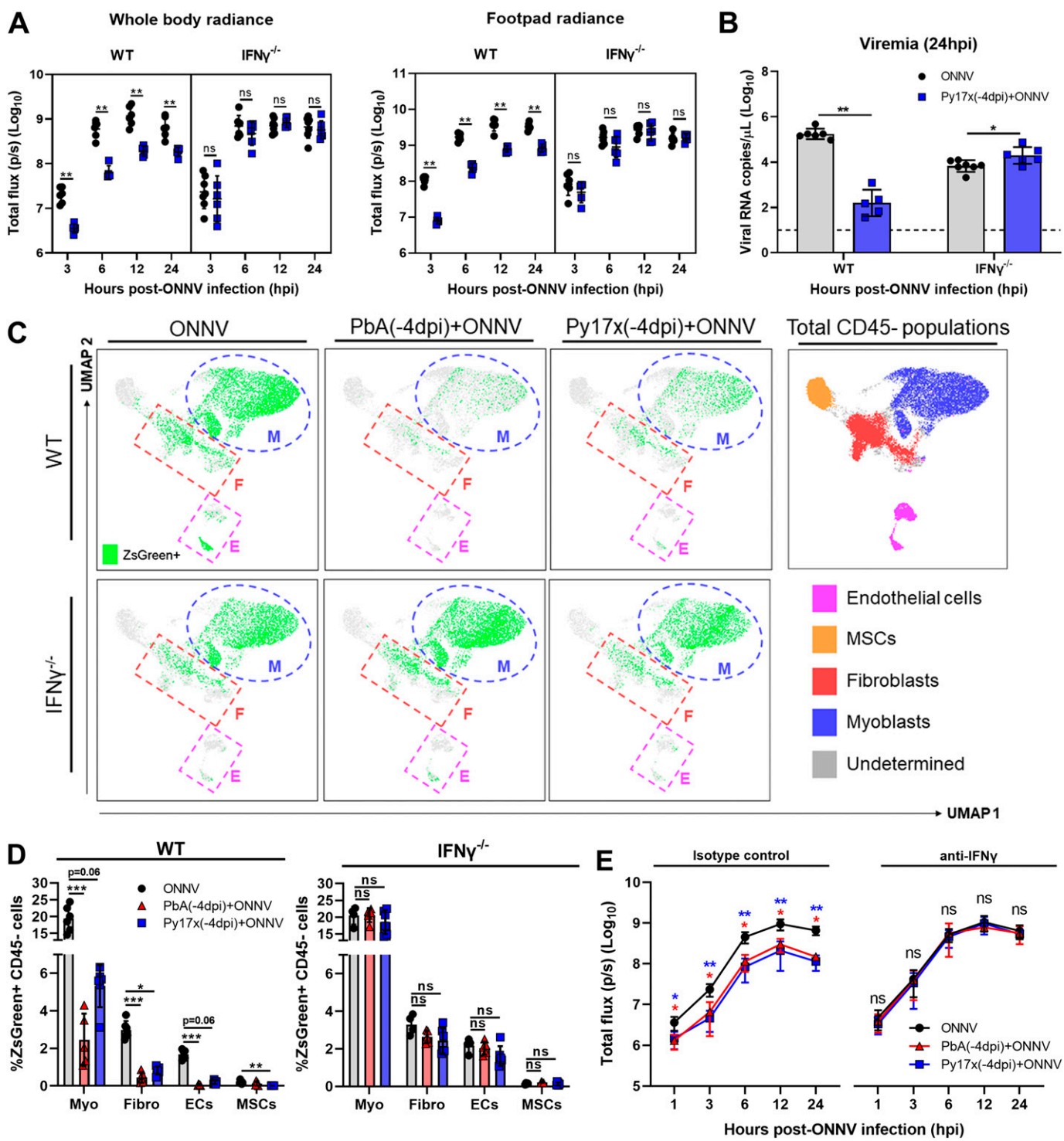

**Figure 5. *Plasmodium*-induced IFNγ mediates the suppression of ONNV replication and dissemination in coinfected animals.**
**(A, B)** In vivo luminescence readings of (A) whole body and footpad radiance at 3, 6, 12, and 24 hpi and (B) viremia at 24 hpi of ONNV and Py17x(-4dpi)+ONNV wild-type (WT), or IFNγ-deficient (IFNγ$^{-/-}$) animals. **(C)** UMAP analysis of 160,000 live CD45$^-$ footpad cells from WT or IFNγ$^{-/-}$ ONNV, PbA(-4dpi)+ONNV, and Py17x(-4dpi)+ONNV groups at 12 hpi. The UMAP plot was generated by concatenation of samples containing 5,000 randomly selected live CD45$^-$ cells from each sample. ONNV, PbA(-4dpi)+ONNV, and Py17x(-4dpi)+ONNV UMAP plots show the global distribution of ZsGreen+ events (ONNV-infected cells). Colored dashed boxes highlight ONNV infection in myoblasts, M; fibroblasts, F and endothelial cells, E. **(D)** Frequency of CD45-ZsGreen+ footpad cells of WT or IFNγ$^{-/-}$ ONNV, PbA(-4dpi)+ONNV, and Py17x(-4dpi)+ONNV groups at 12 hpi. Myo, myoblasts; Fibro, fibroblasts; ECs, endothelial cells and MSCs, mesenchymal stem cells. **(E)** Footpad radiance at 1, 3, 6, 12, and 24 hpi of ONNV, PbA(-4dpi)+ONNV, and Py17x(-4dpi)+ONNV groups in animals treated with mouse anti-IFNγ or isotype control. Data are presented as mean ± SD of at least five animals per experimental group.
**(A, B)** Two-tailed Mann–Whitney U test was used to compute differences between ONNV and Py17x(-4dpi)+ONNV groups in (A) and (B) (*P < 0.05 **P < 0.01). Differences

reduced numbers of ONNV-infected myoblasts, fibroblasts and endothelial cells and this protective effect was reverted in IFNγ-deficient mice or upon in vivo IFNγ neutralization. We therefore hypothesize that production of IFNγ in response to acute blood-stage *Plasmodium* infection could stimulate cells from the CD45⁻ compartment activating antiviral processes (Fig S4) which in turn restrict a subsequent ONNV infection. In line with this, the antiviral effects of IFNγ have been observed in other alphaviruses such as Sindbis virus (SINV). In vitro studies suggested that IFNγ affects SINV replication in mature neurons by interfering with the synthesis of genomic and sub-genomic viral RNA (49) and that this effect is dependent on JAK/STAT signaling (50). The contribution of *Plasmodium*-induced type I IFN (40, 41) to the reduced susceptibility to ONNV infection was also assessed in IFNaR⁻/⁻ mice. Considerable viremia differences were observed between ONNV-infected wild type and IFNaR⁻/⁻ controls (~4–5 $Log_{10}$ at 48 hpi) highlighting the importance of IFN-$\alpha/\beta$ signaling in the control of ONNV infection as observed in other alphavirus animal models (42, 43, 51, 52). Nonetheless, type I IFN-induced upon *Plasmodium* infection seems to be negligible for the establishment of protective effects by murine malaria as coinfected IFNaR⁻/⁻ mice still displayed reduced ONNV infection to a comparable level than coinfected wild-type mice (Fig S6).

It is important to note that in our experiments, mice only experienced suppression of ONNV viremia and virus dissemination after 4 d post-*Plasmodium* inoculation (Figs 1B and 2A–D) and not upon concurrent or sequential (postviral) coinfection. These observations strongly suggested that the timing of parasite inoculation and induction of IFNγ are critical for the protective effects to happen. Interestingly, although the main suppression of ONNV infection occurred in joint footpad cells, we observed lower concentrations of IFNγ in joint footpad tissues compared with serum samples at 4 d post-*Plasmodium* inoculation. Thus, it is likely that IFNγ levels in joint footpads could have increased at an earlier time point. In support of this, we observed high concentrations of IFNγ-induced immune mediators in joint footpad tissues, particularly CXCL10 (53) and CCL7, known to be produced by fibroblasts and mononuclear cells upon IFNγ stimulation (54). The development of T-cell responses, major IFNγ-producing subsets during malaria (ref), could also influence the outcome of a *Plasmodium*–ONNV coinfection in murine models. Early in a blood-stage infection, a large number of IFNγ-secreting Th1 cells are produced, whereas Th2-like responses govern during the chronic phase of infection (17). Because ONNV inoculation occurs in the early stages of murine malaria (4 dpi), it is likely that the antiviral effects of IFNγ are associated to the establishment of Th1 immunity against the parasite. It can be speculated that the degree of virus suppression might differ if ONNV is inoculated during the chronic stage of the murine malaria, particularly when Th1 responses are weaning.

We explored the relevance of our findings in the context of ONNV human infection by treating four human cell lines from different nonimmune lineages (fibroblast, synoviocyte, endothelial, and skeletal muscle cells) with IFNγ before ONNV infection. Mechanistically, our results demonstrated that IFNγ is able of restricting ONNV

infection in human cell lines in a dose dependent manner. Similarly, we also observed that stimulation of skin fibroblasts with plasma from acute *P. vivax*–infected patients containing IFNγ-reduced cell susceptibility to ONNV infection. Of note, the biological effects of IFNγ are mediated conventionally through the activation of the JAK/STAT pathway (55). We postulate that the attenuation of ONNV infection by *Plasmodium*-induced IFN-γ observed in mouse models could be translated to a real coinfection scenario in endemic populations given that the JAK/STAT signaling pathway between humans and mice is highly conserved (56, 57). This suggests that similar downstream effector proteins could be involved in the IFNγ-mediated restriction of alphavirus infection.

A recent study (58) showed that *Plasmodium* infection protected mice from Ebola virus (EBOV)–induced mortality via up-regulation of IFNγ, supporting field reports where coinfected patients by EBOV and *P. falciparum* displayed increased survival rates (59). Conversely, two other murine coinfection models with respiratory viral pathogens such as murine pneumonia virus (PVM) and murine gammaherpesvirus 68 (MHV68) using nonlethal *Plasmodium chabaudi* and *P. yoelii* 17XNL have reported detrimental outcomes for the host such as increased viral loads in the lungs (60) and mortality due to severe anemia (61). These observations were linked to altered type I IFN production (60) and antiviral humoral responses (61) upon coinfection. Thus, the protective or detrimental effects of murine malaria on viral pathogens are likely associated to the modulation of distinct immune responses governing the control of different viral infections. Conversely, other protozoan parasites highly prevalent in the tropics and known to induce the up-regulation of IFNγ in response to infection such as *Leishmania spp.* (62), *Toxoplasma spp.* (63), or *Trypanosoma spp.* (64) could potentially display similar protective effects in coinfection settings. In line with this, mice infected with *Trypanosoma brucei* were shown to be resistant to cutaneous leishmaniasis through the induction of IFNγ generating a hostile pro-inflammatory environment impairing *L. major* colonization of the skin (65).

Results in Fig 1B and E suggest the impairment of ONNV-induced joint pathologies upon *Plasmodium* infection. Interestingly, concurrent coinfection with PbA parasites did not affect viremia levels but significantly reduced the major peak of footpad swelling at 6 dpi. This can be the result of two different mechanisms. First, the development of viremia in concurrently coinfected mice can be attributed to the absence of *Plasmodium*-induced IFNγ in the early stages of ONNV infection in the footpads. It has been shown that the earliest IFNγ production during blood-stage murine malaria only occurs after 24 h postparasite injection (35). During this period, footpad cells in concurrently coinfected mice are still susceptible to ONNV infection which results in viremia levels similar to control mice (Fig 1F). On the other hand, the suppression of joint swelling upon coinfection could be linked to the dysregulation of virus-specific CD4 T-cell responses, main drivers of joint inflammation at 6 dpi (26) by malaria. It has been reported that murine *Plasmodium* infections impair the development of CD4 T-cell responses against heterologous antigens (66, 67). Thus, PbA infections could alter the establishment of virus-specific CD4 T-cell immunity resulting in decreased footpad swelling.

---

between ONNV, PbA(-4dpi)+ONNV, and Py17x(-4dpi)+ONNV groups were calculated using two-tailed Kruskal–Wallis and post hoc Dunn's tests (*$P$ < 0.05 **$P$ < 0.01, ***$P$ < 0.001).

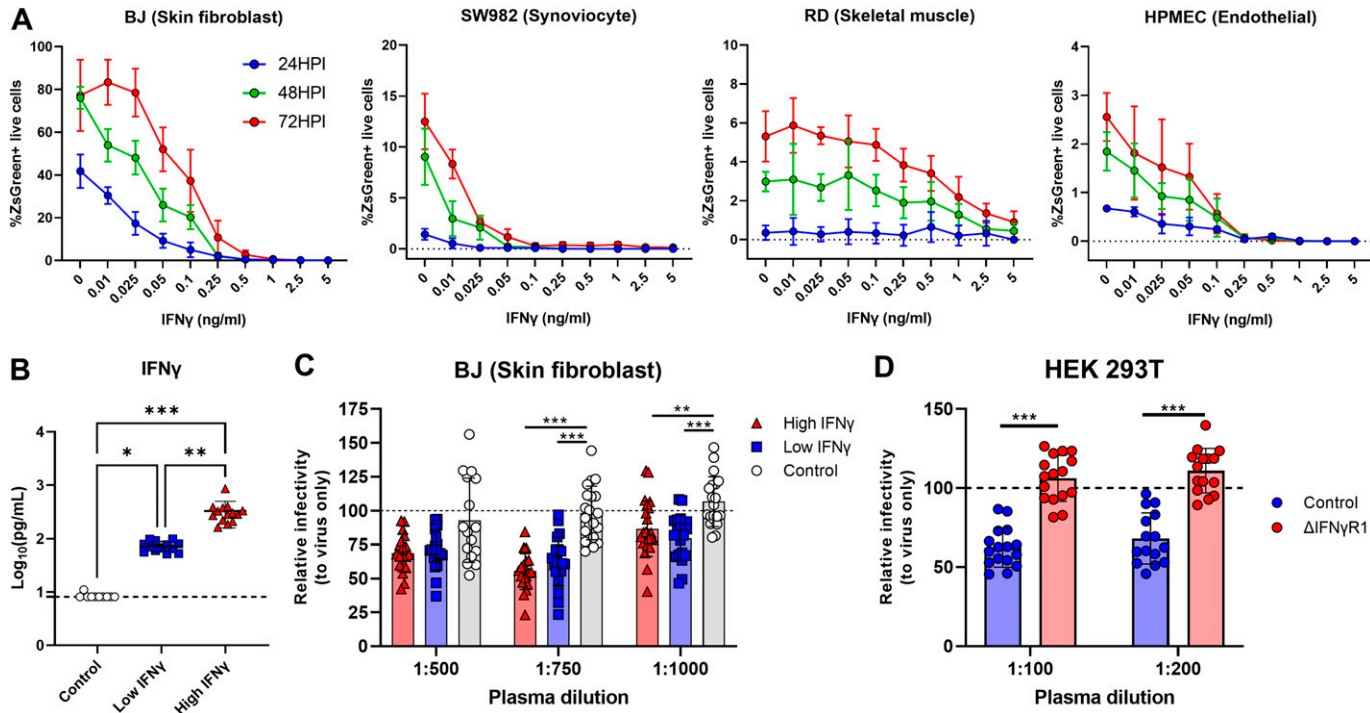

**Figure 6. In vitro stimulation with human IFNγ or plasma from malaria patients reduces susceptibly to ONNV infection.**
For **(A)**, cells were treated with recombinant human IFNγ for 24 h before ONNV infection at MOI 10. **(A)** ONNV infection rates in BJ, SW982, HPMEC, and RD at 24, 48, and 72 hpi. **(B)** IFNγ levels in plasma of healthy controls (HC, n = 10), low (n = 13), and high (n = 14) IFNγ responders. For **(C)**, BJ cells were treated with pooled plasma dilutions from *P. vivax*–infected patients or healthy controls for 12 h before ONNV infection at MOI 1. For **(D)**, control or ΔIFNγR1 HEK293T cells were treated with pooled plasma (1:100 or 1:200) from *P. vivax*-infected patients for 12 h before ONNV infection at MOI 1. Differences between three groups were calculated using two-tailed Kruskal–Wallis and post hoc Dunn's tests, differences between two groups were calculated using two-tailed Mann–Whitney U test (*$P < 0.05$ **$P < 0.01$, ***$P < 0.001$). Data are presented as mean ± SD values and representative of two independent experiments.

The great success of malaria control programs during the last decade has been accompanied of by a sharp rise in the number of arbovirus infections worldwide (68). Our data highlight a possible causative link: that this phenomenon could be due in part to the loss of protective effects exerted by *Plasmodium* infections on alphavirus-induced pathologies that had hitherto masked the real burden of ONNV and other arbovirus infections. This may occur via two mechanisms: first, higher arbovirus case reporting because of more severe symptoms of arbovirus infections in the absence of malaria; second, and worse, increased arbovirus cases might reflect increased arbovirus transmissibility and/or fitness because of increased viral titers in the absence of malaria. Our data in this report will be of value in the fight against *Plasmodium* and ONNV infections in areas where both pathogens co-circulate, particularly highlighting the need for screening and clinical studies of underlying alphavirus infections in malaria intervention programs.

# Materials and Methods

### Mice

3- to 4-wk-old gender-matched wild-type (JAX #000664), IFNγ$^{-/-}$ (JAX #002287), and IFNaR1$^{-/-}$ (JAX #028288) mice in C57BL/6J background were used in this study. Animals were bred and maintained under specific pathogen-free conditions at the Biological Resource Centre of the Agency for Science, Technology, and Research, Singapore (A*STAR).

### Viruses

The IMTSSA/5163 ONNV isolate used in this study was obtained from an acute patient in Chad in 2004 (kindly provided by Marc Grandadam from the Unité de Virologie Tropicale, IMTSSA) (69). Full-length infectious cDNA clones of the IMTSSA/5163 isolate were used to generate ONNV variants expressing the firefly luciferase gene (ONNV-Fluc) and ZsGreen protein (ONNV-ZsGreen) (70, 71). Viruses were propagated in *Aedes albopictus* C6/36 cell line (ATCC CRL-1660) and purified by sucrose-gradient ultracentrifugation. Viral stock titers were determined by standard plaque assay using Vero E6 cells (ATCC CCL-81).

### Parasites

*P. yoelii* 17XNL clone 1.1 (referred as Py17x) was used to induce self-resolving infections in mice (72). Lethal infections were induced by inoculation of *P. berghei* ANKA (PbA) (231cl1) expressing luciferase and GFP under the control of the ef1-α promoter (73). iRBCs were obtained by in vivo serial passage in C57BL/6J mice and were stored in Alsever's solution in liquid nitrogen.

## Human cell lines

BJ (ATCC CRL-2522), SW-982 (ATCC HTB-93), and RD (ATCC CCL-136) cells were grown in DMEM medium with 10% heat-inactivated FBS. HPMEC (#3000; ScienCell) were grown in supplemented EC medium (#1001; ScienCell). Cells were grown at 37°C, relative humidity of 95%, and 5% $CO_2$.

## Generation of ΔIFNγR1 HEK293T cell line

ΔIFNγR1 HEK293T cell line was generated by phosphorylating and annealing primers (5′-CACCGACATGAACCCTATCGTATAT-3′) and (5′-AAACATATACGATA GGGTTCATGTC-3′) (NG_007394.1) using T4 Polynucleotide Kinase (NEB M0201S) in provided buffer supplemented with 1 mM ATP (A2383; Sigma-Aldrich) at 37°C for 30 min followed by 5 min at 95°C and ramped down to 25°C. Annealed primers were then ligated using Instant Sticky-end Ligase Master Mix (M0370; NEB) in pSB-CRIPSR (kindly gifted by Dr Gao and Dr Hu ([74])) previously linearized using Esp3I (R0734S; NEB) in CutSmart Buffer (NEB) and purified using Nucleospin gel and PCR clean up kit (740609.50; Macherey-Nagel). HEK293T cells were co-transfected with pCMV(CAT)T7-SB100 (a gift from Zsuzsanna Izsvak, plasmid # 34879; Addgene; RRID: Addgene_34879 ([75])) and pSB-CRISPR (at 1:1 ratio) using Lipofectamine 2000 (11668019; Thermo Fisher Scientific) in Opti-MEM medium (31985070; Gibco) following the manufacturer's recommendations. 3 d post co-transfection, cells were passaged and cultured in complete media containing 1 μg/ml puromycin (P8833; Sigma-Aldrich). When control cells fully succumbed to puromycin selection, co-transfected cells were cultured in complete media and IFNγR1 expression was assessed by flow cytometry.

## ONNV infection and disease monitoring

Mice were infected with ONNV by subcutaneous inoculation of $10^6$ PFU in 30 μl of PBS in the ventral side of the right hind footpad. Viremia was monitored daily from 1 to 6 dpi and thereafter on every alternate day until 12 dpi. Briefly, 10 μl of blood collected from the tail of each mouse was mixed in 120 μl of PBS supplemented with 10 μl of citrate-phosphate-dextrose solution (Sigma-Aldrich). RNA isolation was performed using the QIAamp Viral RNA kit (QIAGEN) following the manufacturer's instructions with a final elution volume of 60 μl. 1 μl of purified RNA was quantified by qRT-PCR using QuantiTect Probe RT-PCR (QIAGEN) as previously described ([26]). Joint swelling was measured for 2 wk post-ONNV inoculation as a function of height × width relative to measurements preinfection (relative increase) ([26]).

## Tissue viral load determination

Ketamine xylazine–anesthetized mice (150 mg/kg of ketamine, 10 mg/kg of xylazine) were intracardially perfused with PBS and organs were collected in tubes containing zirconia beads (TOMY Digital Biology) and 1 ml TRIzol (Invitrogen) and stored at −80°C. To isolate RNA, tissues were thawed on ice and homogenized using the Bead Ruptor Elite (OMNI International) at a speed of 6 m/s (3 cycles of lysis × 30 s). Tissue lysates were then transferred to 1.5-ml Eppendorf tubes and mixed with 230 μl of chloroform and incubated at RT for 2 min. Samples were centrifuged at 12,000$g$ for 10 min at 4°C and recovered supernatants were transferred into clean Eppendorf Tubes and mixed with 70% ethanol (1:1 volume). RNA was purified using the RNeasy Mini Kit (QIAGEN) according to the manufacturer's protocol. Viral RNA copies were quantified by qRT-PCR as described above.

## In vivo virus tissue dissemination assay

To quantify tissue viral load and virus dissemination in vivo, mice were inoculated with a firefly luciferase-tagged ONNV infectious clone (ONNV-Fluc) and virus dissemination was tracked using the IVIS Spectrum In Vivo Imaging System (Perkin-Elmer) ([26]). Animals were kept anesthetized during the experiment using an oxygen flow rate of 1 liter/minute with 2% isoflurane. Full-body shaved mice were subcutaneously injected with 100 μl of D-luciferin potassium salt (Caliper Life sciences) diluted in PBS (5 mg/ml). Whole body and footpad bioluminescence readings were independently taken 7 min post–D-luciferin injection with a field of view (FOV) of 21.7 cm (ventral position) and 13.1 cm (dorsal position) for whole body (FOV-D) and footpad (FOV-C) measurements, respectively. Two pictures were taken per FOV with exposure times set to "AUTO" and 60 s. Regions of interest were drawn using the software Living Image 3.0 and total flux values (photons/second) were calculated. Readings of naïve mice injected with D-luciferin were used for background subtraction.

## *Plasmodium* infection and disease monitoring

Mice were infected with Py17x or PbA by i.p. injection of $10^6$ iRBC in Alsever's buffer. Parasitemia was monitored by flow cytometry as previously described ([76]) using a staining mix containing anti-mouse APC-tagged CD45 antibodies, 8 μM dihydroethidium (Sigma-Aldrich), and 5 μg/ml Hoechst 33342 (Sigma-Aldrich). Successful infections were confirmed 4 d postinoculation.

## Isolation of footpad cells

Homogenous cell suspensions were obtained for immune profiling of footpads of infected and naïve animals. Mice were culled by cervical dislocation and right paws were harvested and immediately placed in 4 ml of digestion medium containing Collagenase IV (20 μg/ml; Sigma-Aldrich), Dispase I (2 U/ml; Invitrogen), and DNase I (50 μg/ml; Roche Applied Science) mixed in RPMI medium complemented with 10% FBS. Using forceps, footpads were deskinned and deboned to maximize digestion. Processed samples were placed on a shaker and incubated at 37°C at 100 rpm (Biosan PSU-10i) for 3 h. After digestion, tissues were passed through a 40-μm cell strainer (Fisherbrand). Any remaining tissue trapped in the strainer was grinded using the top of a 1 ml syringe plunger to maximize cell recovery. 1× Flow Cytometry Mouse Lysis Buffer (R&D Systems) was used to lyse contaminating RBCs. Samples were resuspended in 1 ml of complete RPMI, overlaid to 35% vol/vol Percoll (Sigma-Aldrich)/RPMI mixture, and centrifuged at 2,400 rpm for 20 min at 4°C. Footpad cell pellets were washed and resuspended in appropriate volumes for counting using haemocytometers.

## Profiling of immune and nonimmune cells by flow cytometry

Cell suspensions were stained for viability using LIVE/DEAD Aqua dye (Life Technologies). Cells were washed and resuspended in 50 $\mu$l of blocking buffer containing TruStain FcX PLUS (anti-mouse CD16/32, clone S17011E) antibody diluted in PBS and incubated in the dark for 15 min on ice. Conjugated anti-mouse antibodies CD45 (30-F11), CD9 (eBioKMC8), CD29 (HM$\beta$1-1), CD31 (390), and Integrin $\alpha$ 7 (334908), Sca-1 (D7) were used to stain cell surface markers for 30 min on ice. Finally, cells were fixed with 50 $\mu$l of eBioscience IC Fixation Buffer (Thermo Fisher Scientific) for 5 min and acquired using a 5-laser LSR II flow cytometer (BD Biosciences) with BD FACSDiva software. Data were analyzed with FlowJo v10.6.2 (Becton, Dickinson and Company).

## Dimensionality reduction analysis of flow cytometry data

Live CD45$^-$ singlets events were pregated and then randomly down-sampled to a fixed number (n = 5,000) for each sample using FlowJo v10.7 (Becton, Dickinson and Company). Down-sampled files were concatenated and analyzed using UMAP for Dimension Reduction plug-in v3.1 using default parameters (number of nearest neighbours = 15, minimum distance = 0.1).

## In vivo IFN$\gamma$ neutralization

Anti-mouse IFN$\gamma$ (0.5 mg per mouse, clone XMG1.2, Bio X Cell) was i.p. injected at 0-, 2-, and 4 d postparasite inoculation. Control groups were given rat IgG1 Isotype control (0.5 mg, clone TNP6A7, Bio X Cell) at similar time points as treatment groups.

## Cytokine/chemokine quantification by multiplexed bead-based immunoassays

Cytokine and chemokine concentrations (pg/ml) were quantified in footpad and serum samples. For footpad samples, animals were anesthetized with ketamine-xylazine and intracardially perfused with PBS. The right paw was cut at the ankle and placed in a gentleMACS M tube (Miltenyi) filled with 1.5 ml of RIPA buffer (50 mM Tris–HCl, pH 7.4, 1% NP-40, 0.25% sodium deoxycholate, 150 mM NaCl, and 1 mM EDTA) complemented with 1× cOmplete Protease Inhibitor Cocktail (Roche). Samples were lysed in a Xiril Dispomix Tissue Homogenizer, centrifuged and the supernatants transferred into clean 2-ml microcentrifuge tubes for sonication in a Branson Ultrasonics Sonifier S-450 (70% intensity × 15 s). For serum samples, blood was collected from the retro-orbicular sinus using a glass Pasteur pipette and allowed to clot for 30 min at room temperature. Clotted blood was centrifuged at 14,000 rpm for serum isolation. Footpad lysates and serum samples were analyzed using the Cytokine & Chemokine 36-Plex Mouse ProcartaPlex Panel 1A (Thermo Fisher Scientific) according to the manufacturer's protocol. Human plasma samples were analyzed using the Cytokine/Chemokine/Growth Factor 45-plex Human ProcartaPlex Panel 1 (Thermo Fisher Scientific). Data were acquired with Luminex FLEXMAP 3D instrument (Millipore) using xPONENT 4.0 software and analyzed with Bio-Plex Manager 6.1.1 (Bio-Rad Laboratories).

## In vitro IFN$\gamma$ treatment and ONNV infection

Recombinant human IFN$\gamma$ (PHC4033; Gibco) diluted in supplemented DMEM at various concentrations was used to treat skin fibroblasts (BJ), synoviocytes (SW-982), skeletal muscle cells (RD), and endothelial cells (HPMEC) for 24 h before virus infection. Cells were washed with PBS and then infected with ONNV-ZsGreen virus at MOI of 10. Cells were harvested at 24-, 48-, and 72-h postinfection (hpi) and percentage of infection was quantified by flow cytometry.

## Data and statistical analyses

Statistical analyses were performed using GraphPad Prism 8.4.3 (GraphPad Software). Data are presented as mean ± SD unless otherwise specified. Nonparametric Mann–Whitney $U$ statistical test was used to compute differences between two groups. Differences between three groups were calculated using two-tailed Kruskal–Wallis and post hoc Dunn's tests. Values obtained for viremia, parasitemia, in vivo imaging, and cytokine/chemokines were log-transformed for representation purposes. $P$-values < 0.05 were considered statistically significant.

## Study approval

Animal experiments were approved by the Institutional Animal Care and Use Committee (IACUC #211635) of A*STAR in accordance with the guidelines of the Agri-Food and Veterinary Authority (AVA) and the National Advisory Committee for Laboratory Animal Research of Singapore (NACLAR). Plasma samples from febrile *P. vivax*–infected patients from Mae Sot, Thailand, were collected and tested in accordance with protocols approved by the University of Oxford Tropical Research Ethics Committee (OXTREC 17-11) and the Ethics Committee of the Faculty of Tropical Medicine at Mahidol University (MUTM 2008-215). Written informed consent was received before participation.

# Supplementary Information

# Acknowledgements

The authors would like to thank Dr Carla Claser for critical discussion and valuable suggestions on the study. We also thank the SIgN Flow Cytometry Core and SIgN Mouse Core for assistance with cytometry analyses and support in animal breeding, respectively. We thank Wilson How from the SIgN Immunomonitoring platform for his support in the multiplexed bead-based immunoassays. We also thank Professor Andres Merits from the University of Tartu for providing the tagged ONNV infectious clones used in this study. The study was supported by a core research grant provided to A*STAR Infectious Diseases Labs and Singapore Immunology Network by the Biomedical Research Council (BMRC) from the Agency for Science, Technology and Research (A*STAR). A Torres-Ruesta is supported by the A*STAR Singapore International Graduate Award (SINGA) scholarship. Flow cytometry platform is supported by the Health and Biomedical Sciences (HBMS) Open Fund Shared Infrastructure Support Grant under the Immunomonitoring Service

Platform project (NRF2017_SISFP09). The funders had no role in the study design, data collection and analysis, decision to publish, or preparation of the manuscript.

## Author Contributions

A Torres-Ruesta: data curation, formal analysis, investigation, visualization, methodology, and writing—original draft, review, and editing.
T-H Teo: data curation, formal analysis, investigation, methodology, and writing—review and editing.
Y-H Chan: formal analysis, investigation, methodology, and writing—review and editing.
SN Amrun: formal analysis, investigation, methodology, and writing—review and editing.
NK-W Yeo: formal analysis, investigation, methodology, and writing—review and editing.
CY-P Lee: formal analysis, investigation, methodology, and writing—review and editing.
SY-T Nguee: formal analysis, investigation, methodology, and writing—review and editing.
MZ Tay: formal analysis, investigation, methodology, and writing—review and editing.
F Nosten: formal analysis, investigation, methodology, and writing—review and editing.
S-W Fong: formal analysis, investigation, methodology, and writing—review and editing.
F-M Lum: formal analysis, investigation, methodology, and writing—review and editing.
G Carissimo: formal analysis, investigation, methodology, and writing—review and editing.
L Renia: conceptualization, supervision, and writing—original draft, review, and editing.
LFP Ng: conceptualization, resources, supervision, funding acquisition, project administration, and writing—original draft, review, and editing.

## Conflict of Interest Statement

The authors declare that they have no conflict of interest.

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
