## [Reviewer comments · Life Science Alliance]

Life Science Alliance

Malaria abrogates O'nyong-nyong virus pathologies by restricting virus infection in non-immune cells

Anthony Torres-Ruesta, Teck-Hui Teo, Yi-Hao Chan, Siti Naqiah Amrun, Nicholas Kim-Wah Yeo, Cheryl Yi Pin Lee, Samantha Yee-Teng Ngeee, Matthew Zirui Tay, Francois Nosten, Siew-Wai Fong, Fok-Moon Lum, Guillaume Carissimo, Laurent Renia, and Lisa Ng

DOI: <https://doi.org/10.26508/lsa.202101272>

Corresponding author(s): Lisa Ng, A*STAR Infectious Diseases Labs and Laurent Renia, A*STAR Infectious Diseases Labs

Review Timeline:

Submission Date:	2021-10-23
Editorial Decision:	2021-11-24
Revision Received:	2021-12-06
Editorial Decision:	2021-12-28
Revision Received:	2022-01-04
Accepted:	2022-01-04

Transaction Report:

November 24, 2021

Re: Life Science Alliance manuscript #LSA-2021-01272-T

Dr. Lisa FP Ng
A*STAR Infectious Diseases Labs

Dear Dr. Ng,

Thank you for submitting your manuscript entitled "Malaria abrogates O'nyong-nyong virus pathologies by restricting virus infection in non-immune cells" to Life Science Alliance. The manuscript was assessed by expert reviewers, whose comments are appended to this letter. We invite you to submit a revised manuscript addressing the Reviewer comments.

Thank you for this interesting contribution to Life Science Alliance. We are looking forward to receiving your revised manuscript.

Sincerely,

B. MANUSCRIPT ORGANIZATION AND FORMATTING:

Reviewer #1 (Comments to the Authors (Required)):

This manuscript by Torres-Ruesta, et al. thoroughly examined pathological outcomes of ONNV infection in the presence / absence of pre-existing Plasmodium infections in murine models. The authors suggested pre-existing Plasmodium blood-stage infections suppress ONNV pathologies and this is dependent on Plasmodium-induced IFN gamma. After treatment with human IFN gamma and plasma samples from *P. vivax* - infected patients, reduced ONNV infection was achieved in different cell lines. An in vitro infection assay using IFN gamma R1-deficient HEK293 cell line further confirmed the conclusion.

There are only a few questions:

1. What is Py and Pb parasitemia in Fig 1B, E, H at the time of monitoring viremia?
2. In Fig 6, different MOIs and different treatment times were used in different conditions. Please explain why.
3. Fig 6D, HEK 293 cells were used to generate IFN gamma R1 deficiency. Is there any specific reason for the cell line used?

Reviewer #2 (Comments to the Authors (Required)):

This article explores the effect of O'nyong-nyong (ONNV) co-infection with malaria, which are often co-circulating and share vectors in Africa. There is prior evidence to suggest that it might be possible to get co-infection but specific mechanistic studies to interrogate this association. In this study, the authors used two different strains of plasmodium to show that pre-infection 4 days prior to viral challenge restricts ONNV replication, dissemination and disease. Timing of co-infection demonstrates that infection at 4 days prior to viral challenge is the most effective but when given at the time of viral infection, the more virulent plasmodium strain PbA protects but the least virulent strain Py17x is not effective. This suggested that an antiviral state is imposed by malarial infection. The authors utilized tagged viruses to follow viral infection by luminescence and flow cytometry. Both demonstrate viral control by prior malaria infection but the flow analysis also had the advantage of demonstrating which cell types were actually infected. Cytokine analysis of serum and footpad lysates from mock vs. PbA and Py17x infected mice suggested that increase in IFN γ production occurs following malaria infection and it was more evident in PbA infected mice relative Py17x which correlates with the increased level of protection from ONNV replication. Infection of IFN γ -/- mice demonstrated equalization of viral infection indicating reduced protection. Treatment with a neutralizing antibody directed against IFN γ validated these findings. The paper finishes by infecting different cell types (endo's, skin fibros, synoviocytes, and skeletal muscle cell lines) with ONNV in the presence of increasing concentrations of IFN γ demonstrating a reduction in viral reporter expression (ZsGreen). Treatment of HEK293 cells with plasma from malarial patients with high or low levels of IFN γ showed a diminishment of viral infectivity that was dependent upon IFN γ R1.

Overall, the paper demonstrates that prior malaria infection impacts the ability of ONNV to replicate and cause disease in mice. There is a link to the presence of IFN γ in mediating viral control. The paper is well-written and accurately describes their findings. Regardless of the effect of malaria on spread, the use of reporter viruses to monitor the site and cell types of virus infection provides interesting descriptive data that has been lacking in the ONNV field. There are a couple of points to address:

1. It is interesting that the more virulent plasmodium strain PbA is more potent at reducing ONNV replication. It is unclear whether this difference is due to increased parasite loads vs. Py17x, wider distribution, or specific tissue targeting. Determining parasite load and tissue distribution data would be helpful to figure this out. If due to loads, then increasing the inoculation dose for Py17x should reproduce the effect. If it is not due to parasite loads then it would be interesting to determine what other factors maybe involved in controlling virus replication.
2. Discussion should be added about why when animals are co-infected simultaneously, they observe a reduction in disease in PbA infected mice but no reduction in viral loads.
3. Results in Figure 5B show that IFN γ -/- mice have a 1-2 log reduction in viral loads in control mice. Is this the predicted outcome? One could argue that viral loads should be higher if IFN γ is limiting viral replication.

Reviewer #3 (Comments to the Authors (Required)):

ONNV, a viral pathogen that is re-emerging in SE Asia, uses the same Anopheline mosquito vector for transmission as does Plasmodium spp. Infecting humans. Therefore, the probability of co-transmission is high and it would be very useful to

understand the dynamics of this situation. The authors have investigated in a mouse model the effects of co-infection with ONNV and *P. yoelii* 17X or *P. berghei* Anka on viremia levels, viral tissue distribution, inflammatory response, and cytokine responses. They further investigate, and make a convincing case for, association of IFN γ specifically with apparent reduction in viremia and tissue involvement. The manuscript is well-written, figures are excellent overall, and there is much to like about the way in which the authors went about establishing the effect of IFN γ . However, the Title, Abstract, and presentation are misleading. The impression is given that malaria generally is protective from ONNV pathology, which is not supported by the data. Moreover, the manuscript generally lacks critique of its own significance.

Major issues

[1] The primary weakness of this manuscript is in fundamental experimental design. The authors investigate a very specific period of malarial infection for its protective ability: the period of maximal host response (and highly elevated IFN γ) to acute asexual blood-stage malarial infection established 4 days prior to viral challenge. No attempt was presented to establish a long-term, persistent infection where IFN γ levels are far less elevated (possible here only with *P. yoelii* 17X) prior to virus introduction, which would be the typical scenario in real life. This is a highly significant shortcoming, because no protective effect was observed when the two infectious agents were given simultaneously, or when the virus was given 4 days prior to Plasmodium. Given the lag involved in malarial hepatic infection prior to entering red blood cells- the phase of infection where IFN γ becomes elevated- the latter situation would more closely mimic the temporal dynamics of true co-transmission infections in a malaria-naïve individual. Thus, while experiments appear to be well-done technically, the results do not mimic natural transmission dynamics in the field, and their significance to co-infection impacts on viral pathology is therefore questionable. This is worthy of critical discussion.

[2] No results were presented regarding the converse side of this interaction, that is, the effects of ONNV infection on the dynamics of malarial growth and survival. Instead, the parasites and the virus were treated as though they were non-replicating reagents rather than significant, independently-acting biological components. This diminishes the broader value and interest of the manuscript.

[3] The primary outcome of the study is a confirmatory, incremental advance only. As malarial infections have long been known to induce elevations in IFN γ , and IFN γ has long been known to have anti-viral effects (the primary outcome of this study), it is unclear what new knowledge was generated or what new insight was gained, especially given the problems of infection timing mentioned above.

[4] Although an effect of adding exogenous IFN γ was demonstrated in *in vitro* culture it would have been relevant to demonstrate this *in vivo*. Throughout, the authors discuss outcomes as though IFN γ acts alone; this should have been tested. The use of IFN γ -/- mice supports the importance of IFN γ , but its absence has effects on several aspects of the development of an otherwise normal immune system. I do not ask that this experiment be done at this point, but this missing bit of the puzzle should at least be a part of the Discussion.

[5] In Supp. Fig. 5 the authors show that the absence of IFN α 1 does not change the qualitative outcome and leave it at that. However, there is a massive quantitative difference that is simply ignored ($1e7$ vs $1e2$ viremias). This needs to be taken into account and explained rather than just accepted as support for their hypothesis. Moreover, the dynamics of viral infection in the WT mice do not agree with those in Figure 1 ($1e2$ vs $1e4$), a result that was not mentioned. These issues require analytical discussion that is lacking.

Minor issues

[1] The authors should reference work establishing the mouse model as one valid for the study of ONNV.

[2] Line 300. "Supp. Fig. 5" should be "Supp. Fig. 6".

[3] Supp. Fig. 6 shows that the effect of exogenous IFN γ plateaus at about 5 ng ml⁻¹ in WT cells. Does this reflect saturation of IFN γ R1? Please explain. Also, characterization of the success of CRISPR/Cas9 "knockdown" has not been provided. It is clear that the overall intensity of the IFN γ R1 signal is reduced, but it remains higher than the isotype background control. It cannot be from a subset of cells with intact receptor genes, or the plot would be bimodal with a small high intensity peak. Please explain the source of this signal.

[4] Line 864. "(A)" should instead be "(C)".

[5] Figure 2C. Please shift the yellow ROI box on the Py17x image slightly to the left. It is partially obscuring observation of the tail.

[6] Figure 4D. It is not obvious how the IFN γ concentration can be so low in the footpads while so high in serum, yet still be responsible for abrogating viral infection and inflammation. Please provide some explanation or discussion.

[7] Figure 6A. Please lower the dotted line indicating 0 for the HPMEC sample to match the other plots, and label the 0 on the Y axis.

Life Science Alliance manuscript #LSA-2021-01272-T entitled "Malaria abrogates O'nyong-nyong virus pathologies by restricting virus infection in non-immune cells"

Reviewer #1 (Comments to the Authors (Required)):

There are only a few questions:

1. What is Py and Pb parasitemia in Fig 1B, E, H at the time of monitoring viremia?

Response: We appreciate the reviewer's comment. Figure R1 reports Py17x and PbA parasitemia in the different co-infection settings during viremia monitoring.

[Figure removed by editorial staff per authors' request]

2. In Fig 6, different MOIs and different treatment times were used in different conditions. Please explain why.

Response: In Figure 6A, the experiments were meant to prove that treatment with recombinant IFN γ exerts antiviral activity in vitro. Given the wide range of IFN γ concentrations tested, we used a high MOI (10) to ensure that almost every cell has the same probability of being infected by at least 1 viral particle (Fields et al., 2007). Pre-treatment with IFN γ for 24 hours was done in accordance to a study by

Burdeinick-Kerr & Griffin (2005) where the antiviral effects of IFN γ against alphavirus Sindbis virus were assessed.

On the other hand, experiments in Figure 6C and 6D were aimed at describing the effects of treatment with plasma from *P. vivax* infected patients in BJ and HEK293T cells. Three reasons led us to use MOI of 1 and 12-hour treatment for these sets of experiments:

i) During optimization experiments, we assessed the earliest incubation period in which plasma samples exerted antiviral activities in BJ cells. 12-hour treatment was enough to observe reduction in ONNV-ZsGreen infection at 1:400, 1:800 and 1:1600 dilutions (Figure R2).

[Figure removed by editorial staff per authors' request]

ii) The IFN γ concentrations in the plasma samples used in these experiments were

in the lower end of the ranges tested in Figure 6A:

- Low IFN γ = 69.53 pg/mL, equivalent to 0.069 ng/mL.
- High IFN γ = 293.675 pg/mL, equivalent to 0.2936 ng/mL.

Given the limitation in plasma availability, pooled samples were further diluted at 1:500, 1:750 and 1:1000 (Figure 6C and 6D). Therefore, using a high MOI (10) could have masked any antiviral effect exerted by IFN γ in vitro. This corroborated the data in Figure R3, which compares the infection profiles of BJ cells treated for 12 hours with plasma samples diluted at 1:400, 1:800 and 1:1600 and infected with ONNV-ZsGreen at MOI of 1 or MOI of 10.

[Figure removed by editorial staff per authors' request]

P.vivax-infected patients (n=14, high IFN γ responders) diluted at 1:400, 1:800 or 1:1600 for 12 hours prior to ONNV-ZsGreen infection at MOI of 1 or 10. Percent infection was calculated relative to virus only wells (dotted line). Data are presented as mean \pm SD values.

iii) Given that HEK293T cells are highly permissive to alphavirus infection (Wikan et al., 2012), we used a low MOI (1) to avoid high cell mortality by the time of harvest (24 hpi) (Figure R4). We decided to match this MOI (1) for experiments involving plasma treatment with BJ and HEK293 T cells (Figure 6C and 6D).

[Figure removed by editorial staff per authors' request]

References:

- Fields BN, Knipe DM, Howley PM. Fields virology: Part 1, 5th Ed. Philadelphia: Wolters Kluwer Health/Lippincott Williams & Wilkins. 2007.
- Burdeinick-Kerr R, Griffin DE. Gamma interferon-dependent, noncytolytic clearance of sindbis virus infection from neurons in vitro. J Virol. 2005 May;79(9):5374-85.
- Wikan N, Sakoonwatanyoo P, Ubol S, Yoksan S, Smith DR. Chikungunya virus infection of cell lines: analysis of the East, Central and South African lineage. PLoS One. 2012;7(1):e31102..

3. Fig 6D, HEK 293 cells were used to generate IFN gamma R1 deficiency. Is there any specific reason for the cell line used?

Response: We thank the reviewer for this comment. We decided to use HEK293T cells given their high permissivity and stability for transfection and fast doubling time of ~24 hours (Graham et al, 1977; Thomas & Smart, 2005) compared to 72 to 80 hours in BJ cells (Pereira da Silva et al., 2014). This characteristic was key when assessing the stability of IFNgR1-deficiency in HEK293T cells for up to 8 passages (Supp. Fig. 6) prior to experiments in Figure 6D .

References:

- Graham FL, Smiley J, Russell WC, Nairn R. Characteristics of a human cell line transformed by DNA from human adenovirus type 5. *J Gen Virol.* 1977 Jul;36(1):59-74.
- Thomas P, Smart TG. HEK293 cell line: a vehicle for the expression of recombinant proteins. *J Pharmacol Toxicol Methods.* 2005 May-Jun;51(3):187-200.
- Pereira da Silva L, Miguel Neves B, Moura L, Cruz MT, Carvalho E. Neurotensin decreases the proinflammatory status of human skin fibroblasts and increases epidermal growth factor expression. *Int J Inflam.* 2014;2014:248240.

Reviewer #2 (Comments to the Authors (Required)):

1. It is interesting that the more virulent plasmodium strain PbA is more potent at reducing ONNV replication. It is unclear whether this difference is due to increased parasite loads vs. Py17x, wider distribution, or specific tissue targeting. Determining parasite load and tissue distribution data would be helpful to figure this out. If due to loads, then increasing the inoculation dose for Py17x should reproduce the effect. If it is not due to parasite loads then it would be interesting to determine what other factors maybe involved in controlling virus replication.

Response: The reviewer raised a good point. The parasite loads upon sequential co-infection were indeed measured and are shown in Figure R5. At the time of virus inoculation (4 days post-parasite infection), the parasite burden in mice infected with Py17x was higher compared to animals infected with PbA. However, PbA infection induced higher IFN γ levels in serum and footpad tissues at 4dpi compared to Py17x (Figure 4D). It is likely that acute PbA infections are more potent at limiting ONNV burden due to the induction of stronger IFN γ responses than Py17x. This is also supported by observations of murine malaria in 3-week-old C57BL/6/J mice where IFN γ levels in supernatants of cultured splenocytes from infected mice (5dpi) were higher upon infection with PbA (~1000pg/mL) than with Py17x (~500pg/mL) (Shan et al., 2012; Shan et al., 2013).

[Figure removed by editorial staff per authors' request]

References:

- Shan Y, Liu J, Jiang YJ, Shang H, Jiang D, Cao YM. Age-related susceptibility and resistance to nonlethal *Plasmodium yoelii* infection in C57BL/6 mice. *Folia Parasitol (Praha)*. 2012 Sep;59(3):153-61.
- Shan Y, Liu J, Pan YY, Jiang YJ, Shang H, Cao YM. Age-related CD4(+)CD25(+)Foxp3(+) regulatory T-cell responses during *Plasmodium berghei* ANKA infection in mice susceptible or resistant to cerebral malaria. *Korean J Parasitol*. 2013 Jun;51(3):289-95.

2. Discussion should be added about why when animals are co-infected simultaneously, they observe a reduction in disease in PbA infected mice but no reduction in viral loads.

Response: We thank the reviewer for raising this point and to improve our manuscript, we have now included the following paragraph (lines 372-387) in the discussion section:

“Results in Figure 1B and 1E suggest the impairment of ONNV-induced joint pathologies upon *Plasmodium* infection. Interestingly, concurrent co-infection with PbA parasites did not affect viremia levels but significantly reduced the major peak of footpad swelling at 6 dpi. This can be the result of two different mechanisms. First, the development of viremia in concurrently co-infected mice can be attributed to the absence of *Plasmodium*-induced IFN γ in the early stages of ONNV infection in the footpads. It has been shown that the earliest IFN γ production during blood-stage murine malaria only occurs after 24 hours post-parasite injection (De Souza et al., 1997). During this period, footpad cells in concurrently co-infected mice are still susceptible to ONNV infection which results in viremia levels similar to control mice (Figure 1F). On the other hand, the suppression of joint swelling upon co-infection could be linked to the dysregulation of virus-specific CD4 T cell responses, main drivers of joint inflammation at 6 dpi (Chan et al., 2020) by malaria. It has been reported that murine *Plasmodium* infections impair the development of CD4 T cell responses against heterologous antigens (Lucas et al., 1996; Luyendyk et al., 2002; Millington et al., 2006). Thus, PbA infections could alter the establishment of virus-specific CD4 T cell immunity resulting in decreased footpad swelling”.

References:

- De Souza JB, Williamson KH, Otani T, Playfair JH. Early gamma interferon responses in lethal and nonlethal murine blood-stage malaria. *Infect Immun.* 1997 May;65(5):1593-8. doi: 10.1128/iai.65.5.1593-1598.1997.
- Chan YH, Teo TH, Torres-Ruesta A, Hartimath SV, Chee RS, Khanapur S, Yong FF, Ramasamy B, Cheng P, Rajarethinam R, Robins EG, Goggi JL, Lum FM, Carissimo G, Rénia L, Ng LFP. Longitudinal [18F]FB-IL-2 PET Imaging to Assess the Immunopathogenicity of O'nyong-nyong Virus Infection. *Front Immunol.* 2020 May 12;11:894.
- Lucas B, Kasper LH, Smith K, Haque A. In vivo treatment with interleukin 2 reduces parasitemia and restores IFN-gamma gene expression and T-cell proliferation during acute murine malaria. *C R Acad Sci III.* 1996; 319:8.
- Luyendyk J, Olivas OR, Ginger LA, Avery AC. Antigen-presenting cell function during *Plasmodium yoelii* infection. *Infect Immun.* 2002 Jun;70(6):2941-9.
- Millington OR, Di Lorenzo C, Phillips RS, Garside P, Brewer JM. Suppression of adaptive immunity to heterologous antigens during *Plasmodium* infection through hemozoin-induced failure of dendritic cell function. *J Biol.* 2006;5(2):5.

3. Results in Figure 5B show that IFN γ ^{-/-} mice have a 1-2 log reduction in viral loads in control mice. Is this the predicted outcome? One could argue that viral loads should be higher if IFN γ is limiting viral replication.

Response: We note the reviewer's concern. As pointed out by Reviewer 3 (Major concerns #3), the absence of IFN γ in genetically modified mice could affect various aspects of the development of an otherwise normal immune system. We attribute

this 1-2 log “reduction” in viral loads of control mice to innate differences between WT and IFN γ ^{-/-} animals. For fairer comparison, we have included viremia levels and footpad viral loads of co-infected wild-type mice treated with anti-IFN γ or isotype control prior ONNV infection (Figure R6). Viremia (Figure R6A) and footpad viral load profiles (Figure R6B) were similar in control mice treated with anti-IFN γ or isotype antibodies.

[Figure removed by editorial staff per authors' request]

Reviewer #3 (Comments to the Authors (Required)):

Major issues

[1] The primary weakness of this manuscript is in fundamental experimental design. The authors investigate a very specific period of malarial infection for its protective ability: the period of maximal host response (and highly elevated IFN γ) to acute asexual blood-stage malarial infection established 4 days prior to viral challenge. No attempt was presented to establish a long-term, persistent infection where IFN γ levels are far less elevated (possible here only with *P. yoelii* 17X) prior to virus introduction, which would be the typical scenario in real life. This is a highly significant shortcoming, because no protective effect was observed when the two infectious agents were given simultaneously, or when the virus was given 4 days prior to Plasmodium. Given the lag involved in malarial hepatic infection prior to entering red blood cells- the phase of infection where IFN γ becomes elevated- the latter situation would more closely mimic the temporal dynamics of true co-transmission infections in a malaria-naïve individual. Thus, while experiments appear to be well-done technically, the results do not mimic natural transmission dynamics in the field, and their significance to co-infection impacts on viral pathology is therefore questionable. This is worthy of critical discussion.

Response: We thank the reviewer the comments and we are aware of the limitations of the study. However, we do not agree that there is a problem with the design. Nonetheless, we note the concerns and to improve clarity, we have included a new paragraph (332-345) in the discussion to further explain the various findings:

“It is important to note that in our experiments, mice only experienced suppression of ONNV viremia and virus dissemination after 4 days post-Plasmodium inoculation (Figure 1B and Figure 2A-D) and not upon concurrent or sequential (post-viral) co-infection. These observations strongly suggested that the timing of parasite inoculation and induction of IFN γ are critical for the protective effects to happen. It has been described that murine malaria infections involve a biphasic activation of helper T cell responses (Perez-Mazliah & Langhorne, 2014). Early in blood-stage infection, a large number of IFN γ -secreting Th1 cells are produced, while Th2-like responses govern during the chronic phase of infection (Kurup et al., 2019). Since ONNV inoculation occurs in the early stages of murine malaria (4 dpi), it is likely that the antiviral effects of IFN γ are associated to the establishment of Th1 immunity against the parasite. It can be speculated that the degree of virus suppression might differ if ONNV is inoculated during the chronic stage of the murine malaria, particularly when Th1 responses are waning.”

References:

- Perez-Mazliah D, Langhorne J. CD4 T-cell subsets in malaria: TH1/TH2 revisited. *Front Immunol.* 2015 Jan 12;5:671. doi: 10.3389/fimmu.2014.00671. PMID: 25628621; PMCID: PMC4290673.
- Kurup SP, Butler NS, Harty JT. T cell-mediated immunity to malaria. *Nat Rev Immunol.* 2019 Jul;19(7):457-471. doi: 10.1038/s41577-019-0158-z. PMID: 30940932; PMCID: PMC6599480.

[2] No results were presented regarding the converse side of this interaction, that is, the effects of ONNV infection on the dynamics of malarial growth and survival. Instead, the parasites and the virus were treated as though they were non-replicating reagents rather than significant, independently-acting biological components. This diminishes the broader value and interest of the manuscript.

Response: We appreciate the reviewer's comment. The following lines report the effects of ONNV infection on the dynamics of malarial growth and survival:

The inoculation of ONNV four days post Py17x or PbA infection did not alter parasitemia development (Figure R7A and R7B) or PbA-induced mortality (Figure R7B).

Concurrent co-infection with non-lethal Py17x and ONNV resulted in increased parasitemia levels (Figure R7C). Py17x parasitemia resolution was delayed as co-infected animals took 26-28 days to clear blood-stage parasites compared to 20-22 days in controls. On the other hand, simultaneous inoculation with PbA and ONNV did not affect the development of parasitemia or ECM mortality in co-infected mice (Figure R7D).

The infection with Plasmodium parasites four days post ONNV inoculation aggravated Py17x and PbA parasitemia (Figure R7E and R7F) but did not impact ECM mortality (Figure R7F).

We have included these results here for review purposes.

[Figure removed by editorial staff per authors' request]

[3] The primary outcome of the study is a confirmatory, incremental advance only. As malarial infections have long been known to induce elevations in IFN γ , and IFN γ has long been known to have anti-viral effects (the primary outcome of this study), it is unclear what new knowledge was generated or what new insight was gained, especially given the problems of infection timing mentioned above.

Response: We appreciate the reviewer's comment. It is true that human and animal malaria infections trigger IFN γ production during the acute phase of the disease. Two other murine co-infection models with respiratory viral pathogens such as murine pneumonia virus (PVM) and murine gammaherpesvirus 68 (MHV68) using non-lethal *P. chabaudi* and *P. yoelii* 17XNL have reported detrimental outcomes for the host such as increased viral loads in the lungs and mortality due to severe anaemia (Edwards et al., 2015; Matar et al., 2015).

Importantly, our work highlights that immune effector mechanisms triggered by acute *Plasmodium* infections modulate the behaviour of non-immune cell subsets that are main ONNV targets. While, it is not a major breakthrough, this work paves the way for further studies in understanding interactions between malaria and arboviral diseases in endemic areas, which remains important.

References:

- Edwards CL, Zhang V, Werder RB, Best SE, Sebina I, James KR, Faleiro RJ, de Labastida Rivera F, Amante FH, Engwerda CR, Phipps S, Haque A. Coinfection with Blood-Stage Plasmodium Promotes Systemic Type I Interferon Production during Pneumovirus Infection but Impairs Inflammation and Viral Control in the Lung. *Clin Vaccine Immunol*. 2015 May;22(5):477-83.
- Matar CG, Anthony NR, O'Flaherty BM, Jacobs NT, Priyamvada L, Engwerda CR, Speck SH, Lamb TJ. Gammaherpesvirus Co-infection with Malaria Suppresses Anti-parasitic Humoral Immunity. *PLoS Pathog*. 2015 May 21;11(5):e1004858.

[4] Although an effect of adding exogenous IFN γ was demonstrated in in vitro culture it would have been relevant to demonstrate this in vivo. Throughout, the authors discuss outcomes as though IFN γ acts alone; this should have been tested. The use of IFN γ -/- mice supports the importance of IFN γ , but its absence has effects on several aspects of the development of an otherwise normal immune system. I do not ask that this experiment be done at this point, but this missing bit of the puzzle should at least be a part of the Discussion.

Response: We appreciate the reviewer's comment. Indeed, to confirm the observations from IFN γ -deficient mice, we conducted an in-vivo IFN γ neutralization experiment using anti-IFN γ antibodies, which resulted in the reestablishment of footpad viral load levels in co-infected animals to a similar degree to ONNV controls (Figure 5E). This suggests that the IFN γ alone is the main cytokine driving the antiviral effects exerted; however, we do not belittle the contribution of other cytokine

and chemokines produced by *Plasmodium* infection in the establishment of an antiviral status in mouse tissues.

[5] In Supp. Fig. 5 the authors show that the absence of IFN α R1 does not change the qualitative outcome and leave it at that. However, there is a massive quantitative difference that is simply ignored (1e7 vs 1e2 viremias). This needs to be taken into account and explained rather than just accepted as support for their hypothesis. Moreover, the dynamics of viral infection in the WT mice do not agree with those in Figure 1 (1e2 vs 1e4), a result that was not mentioned. These issues require analytical discussion that is lacking.

Response:We thank the reviewer for the comment. In fact, it is not surprising that IFN α R1^{-/-} deficient animals infected with ONNV develop markedly high viremia levels compared to wild type (WT) mice (Supp. Fig. 5). Previous studies using experimental infections of type I IFN-deficient mice with ONNV (Seymour et al., 2013) or related alphaviruses such as CHIKV (Couderc et al., 2008; Schilte et al., 2010) or SINV have reported similar findings with even greater viral load differences between WT animals and IFN α R1^{-/-} which ultimately resulted in death. Of note, a study from Couderc et al. (2008) assessed the viral load differences between WT and IFN α R1^{-/-} upon the administration of 10E6 PFU intradermally in the ventral thorax. At 3 dpi, IFN α R1^{-/-} mice displayed 5-6 Log₁₀ higher viral load (TCID₅₀) in the serum, liver, spleen, muscle and joints compared to WT mice.

[Figure removed by editorial staff per authors' request]

Regarding the variability observed in the dynamics of ONNV infection in WT mice between experiments conducted for Figure 1 and Supp. Figure 5; this might be due to batch-to-batch differences in the animals used. Indeed, it is not uncommon to

observed variability in the viral load trends between experiments. These observations have been also reported in other published studies. In a study by Nguyen et al. (2020), footpad injection of ONNV isolate IMTSSA/2004/5163 (same strain used for our work) in C57BL6/J mice resulted in variable viremia levels across four independent experiments.

[Figure removed by editorial staff per authors' request]

We believe that the conclusions generated by our experiments using IFN α 1^{-/-} deficient animals regarding the involvement of type I IFN responses in the antiviral effects exerted by *Plasmodium*-induced IFN γ are valid. In fact, they are also supported by data generated by Rogers et al. (2020), where type I IFN responses were not essential for the establishment of *Plasmodium*-induced protection against a recombinant version of Ebola virus in mice.

References:

- Seymour RL, Rossi SL, Bergren NA, Plante KS, Weaver SC. The role of innate versus adaptive immune responses in a mouse model of O'nyong-nyong virus infection. *Am J Trop Med Hyg.* 2013 Jun;88(6):1170-9.
- Couderc T, Chrétien F, Schilte C, Disson O, Brigitte M, Guivel-Benhassine F, Touret Y, Barau G, Cayet N, Schuffenecker I, Desprès P, Arenzana-Seisdedos F, Michault A, Albert ML, Lecuit M. A mouse model for Chikungunya: young age and inefficient type-I interferon signaling are risk factors for severe disease. *PLoS Pathog.* 2008 Feb 8;4(2):e29.
- Schilte C, Couderc T, Chretien F, Sourisseau M, Gangneux N, Guivel-Benhassine F, Kraxner A, Tschopp J, Higgs S, Michault A, Arenzana-Seisdedos F, Colonna M, Peduto L, Schwartz O, Lecuit M, Albert ML. Type I IFN controls chikungunya virus via its action on nonhematopoietic cells. *J Exp Med.* 2010 Feb 15;207(2):429-42. doi: 10.1084/jem.20090851.
- Ryman KD, Klimstra WB, Nguyen KB, Biron CA, Johnston RE. Alpha/beta interferon protects adult mice from fatal Sindbis virus infection and is an important determinant of cell and tissue tropism. *J Virol.* 2000 Apr;74(7):3366-78. doi: 10.1128/jvi.74.7.3366-3378.2000.
- Nguyen W, Nakayama E, Yan K, Tang B, Le TT, Liu L, Cooper TH, Hayball JD, Faddy HM, Warrilow D, Allcock RJN, Hobson-Peters J, Hall RA, Rawle DJ, Lutzky VP, Young P, Oliveira NM, Hartel G, Howley PM, Prow NA, Suhrbier A. Arthritogenic Alphavirus Vaccines: Serogrouping Versus Cross-Protection in Mouse Models. *Vaccines (Basel).* 2020 May 5;8(2):209.
- Rogers KJ, Shtanko O, Vijay R, Mallinger LN, Joyner CJ, Galinski MR, Butler NS, Maury W. Acute Plasmodium Infection Promotes Interferon-Gamma-Dependent Resistance to Ebola Virus Infection. *Cell Rep.* 2020 Mar 24;30(12):4041-4051.e4.

Minor issues

[1] The authors should reference work establishing the mouse model as one valid for the study of ONNV.

Response: We referenced a previously published work on the establishment of an immunocompetent mouse model of ONNV in the original manuscript. Lines 102-104 read as follows:

“An immunocompetent mouse model was previously established to recapitulate ONNV-induced joint pathologies (inflammation, edema, muscle necrosis, synovitis, and tenosynovitis) and acute viremia (Chan et al, 2019).”

Reference:

- Chan YH, Teo TH, Torres-Ruesta A, Hartimath SV, Chee RS, Khanapur S, Yong FF, Ramasamy B, Cheng P, Rajarethinam R, Robins EG, Goggi JL, Lum FM, Carissimo G, Rénia L, Ng LFP. Longitudinal [18F]FB-IL-2 PET Imaging to Assess the Immunopathogenicity of O'nyong-nyong Virus Infection. *Front Immunol.* 2020 May 12;11:894.

[2] Line 300. "Supp. Fig. 5" should be "Supp. Fig. 6".

Response: We thank the reviewer for pointing out this oversight. We have updated line 297 to read: “we generated a HEK293T cell line with impaired IFN γ signalling (Supplementary Figure 6)”.

[3] Supp. Fig. 6 shows that the effect of exogenous IFN γ plateaus at about 5 ng ml⁻¹ in WT cells. Does this reflect saturation of IFN γ R1? Please explain. Also, characterization of the success of CRISPR/Cas9 “knockdown” has not been provided. It is clear that the overall intensity of the IFN γ R1 signal is reduced, but it remains higher than the isotype background control. It cannot be from a subset of cells with intact receptor genes, or the plot would be bimodal with a small high intensity peak. Please explain the source of this signal.

Response:

We thank the reviewer for the comment. Indeed, the expression level of IFN γ R in HEK293T cells is low compared to other epithelial cell lines such as HeLa cells (Burova et al, 2007). We hypothesize that the plateau observed upon stimulation with IFN γ concentrations >5ng/ml is likely due to the saturation of limited IFN γ R units in HEK293T cells.

Regarding the characterization of Δ IFN γ R1 HEK293T cells, we have now included the profiles of IFN γ R1 expression in HEK293T cells post CRISPR/Cas9 “knockdown” (Pre-sorting) and expansion of sorted IFN γ R1-deficient HEK293T for up to eight passages as part of Supp. Fig. 6 . These results suggest that the loss of IFN γ R1 is stable in Δ IFN γ R1 HEK293T cells.

Upon close examination, we realized that the “higher” IFN γ R1 expression in Δ IFN γ R1 HEK293T cells compared to the isotype background control was the result of mislabelling. In fact, the signal from isotype background controls was “higher” in Δ IFN γ R1 HEK293T, which was observed as a slight shift of signal to the right (e.g. passage 7 and 8). This might be due to non-specific binding of the tagged-isotype control antibody or differences in the fluorochrome conjugation between isotype control and anti-IFN γ R1 antibodies. Nevertheless, these trivial differences do not impact the loss of responsiveness to IFN γ in Δ IFN γ R1 HEK293T cells, as IFN γ -stimulated Δ IFN γ R1 HEK293T cells have similar infection profiles than untreated control HEK293T cells (Supp. Fig. 6B,).

References:

- Burova E, Vassilenko K, Dorosh V, Gonchar I, Nikolsky N. Interferon gamma-dependent transactivation of epidermal growth factor receptor. *FEBS Lett.* 2007 Apr 3;581(7):1475-80. doi: 10.1016/j.febslet.2007.03.002. Epub 2007 Mar 8. PMID: 17362940.

[Figure removed by editorial staff per authors' request]

[4] Line 864. "(A)" should instead be "(C)".

Response: We thank the reviewer for pointing out this oversight. We have updated line 855 to read: "(C) Principal component analysis using differentially expressed..."

[5] Figure 2C. Please shift the yellow ROI box on the Py17x image slightly to the left. It is partially obscuring observation of the tail.

Response: We have updated Figure 2C accordingly so that the yellow ROI box do not mast the observation of the tail.

[Figure removed by editorial staff per authors' request]

[6] Figure 4D. It is not obvious how the IFN γ concentration can be so low in the footpads while so high in serum, yet still be responsible for abrogating viral infection and inflammation. Please provide some explanation or discussion.

Response: Indeed, it is likely that the kinetics of IFN γ production in the serum and footpad tissues upon *Plasmodium* infection are different. Unfortunately, we only quantified IFN γ concentrations at 4 dpi; thus, it is possible that IFN γ levels in footpad tissue could have increased at an earlier time point. In support of this, we observed

higher concentrations of IFN γ -induced cytokines and chemokines in footpad tissues, particularly IFN γ -induced CXCL10 (Metzemaekers et al., 2018), which was found to be elevated by ~20- and ~11-folds in footpad lysates of PbA and Py17x-infected animals. Similarly, CCL7, known to be produced by fibroblasts and mononuclear cells upon IFN γ stimulation (Liu et al., 2018), was found to be elevated by ~8- and ~4-folds in footpad lysates of PbA and Py17x-infected animals.

References:

- Metzemaekers M, Vanheule V, Janssens R, Struyf S, Proost P. Overview of the Mechanisms that May Contribute to the Non-Redundant Activities of Interferon-Inducible CXC Chemokine Receptor 3 Ligands. *Front Immunol.* 2018 Jan 15;8:1970.
- Liu Y, Cai Y, Liu L, Wu Y, Xiong X. Crucial biological functions of CCL7 in cancer. *PeerJ.* 2018 Jun 14;6:e4928.

[7] Figure 6A. Please lower the dotted line indicating 0 for the HPMEC sample to match the other plots, and label the 0 on the Y axis.

Response: We have updated Figure 6A accordingly. Dotted line for ONNV infection of HPMECs now matches the other plots and the label 0 has been added on the Y-axis.

December 28, 2021

RE: Life Science Alliance Manuscript #LSA-2021-01272-TR

Dr. Lisa FP Ng
A*STAR Infectious Diseases Labs
8A Biomedical Grove
Singapore 138665

Dear Dr. Ng,

Thank you for submitting your revised manuscript entitled "Malaria abrogates O'nyong-nyong virus pathologies by restricting virus infection in non-immune cells". We would be happy to publish your paper in Life Science Alliance pending final revisions necessary to meet our formatting guidelines. Please address Reviewer 3's remaining comments.

- please add ORCID ID for the corresponding (and secondary corresponding) author-you both should have received instructions on how to do so
- we encourage you to revise the figure legends for Figure 1 such that the figure panels are introduced in alphabetical order
- please add callouts for Figure S2C, D to your main manuscript text

A. FINAL FILES:

B. MANUSCRIPT ORGANIZATION AND FORMATTING:

****It is Life Science Alliance policy that if requested, original data images must be made available to the editors. Failure to provide**

original images upon request will result in unavoidable delays in publication. Please ensure that you have access to all original data images prior to final submission.**

The license to publish form must be signed before your manuscript can be sent to production. A link to the electronic license to publish form will be sent to the corresponding author only. Please take a moment to check your funder requirements.

Sincerely,

Reviewer #1 (Comments to the Authors (Required)):

The revised version excellently answered my points. I have no more comments.

Reviewer #2 (Comments to the Authors (Required)):

The authors have adequately addressed my specific comments and concerns.

Reviewer #3 (Comments to the Authors (Required)):

This revised manuscript is improved over the original submission. The authors are thanked for being responsive to the criticisms raised by the reviewers. My apologies for two criticisms related to information that was overlooked in the original. Mostly my legitimate concerns were adequately addressed, with the exceptions below.

Major issues:

original point 5: I do not find the authors' response satisfactory. Not only is there a massive quantitative difference in viremia (4-5 logs) between wt and IFN α 1 $^{-/-}$, it is still increasing at 48h in the knockouts. Clearly, the presence or absence of IFN α 1 has a profound effect on control of viral replication that the authors simply do not address. This should be explained as well as possible in the Discussion, including how it impacts their conclusions regarding IFN γ .

Minor issues:

original point 2: The authors should consider including the supplementary data presented in the rebuttal regarding effects on malarial populations in the Supplementary Results.

original point 3: The authors' response regarding distinctions in immune responses to different viruses (and supporting reference) should be incorporated in the Discussion.

original point 6: The authors' response should be incorporated into the Discussion. There are many dynamics to this system that are not obvious, and the apparent disconnect between circulating and tissue cytokine levels and how the two relate to immune protection is one of them.

Life Science Alliance manuscript #LSA-2021-01272-T entitled "Malaria abrogates O'nyong-nyong virus pathologies by restricting virus infection in non-immune cells"

Reviewer #1 (Comments to the Authors (Required)):

The revised version excellently answered my points. I have no more comments.

Reviewer #2 (Comments to the Authors (Required)):

The authors have adequately addressed my specific comments and concerns.

Reviewer #3 (Comments to the Authors (Required)):

This revised manuscript is improved over the original submission. The authors are thanked for being responsive to the criticisms raised by the reviewers. My apologies for two criticisms related to information that was overlooked in the original. Mostly my legitimate concerns were adequately addressed, with the exceptions below.

Major issues:

original point 5: I do not find the authors' response satisfactory. Not only is there a massive quantitative difference in viremia (4-5 logs) between wt and IFNaR1^{-/-}, it is still increasing at 48h in the knockouts. Clearly, the presence or absence of IFNaR1 has a profound effect on control of viral replication that the authors simply do not address. This should be explained as well as possible in the Discussion, including how it impacts their conclusions regarding IFN γ .

Response: We thank the reviewer for the comment. We note the concerns and to improve clarity, we have edited the paragraph (lines 278-285) in the results section describing the effects of co-infection in IFNaR1^{-/-} mice.

"To evaluate any possible contribution of type I interferon responses in the reduced susceptibility to ONNV upon co-infection, the effect of pre-existing murine malaria on ONNV replication was assessed in IFNaR^{-/-} mice (deficient of interferon- α/β receptor). Viremia measurements at 12, 24 and 48 hpi in co-infected IFNaR^{-/-} mice (Fig S5) revealed that murine malaria was still able to restrict ONNV infection, ruling out the involvement of type I interferon responses in the antiviral effects exerted by *Plasmodium*-induced IFN γ ".

Similarly, we have updated the Discussion section as follows (lines 340-348).

"The contribution of *Plasmodium*-induced type I interferon [40, 41] to the reduced susceptibility to ONNV infection was also assessed in IFNaR^{-/-} mice. Considerable viremia differences were observed between ONNV-infected wild type and IFNaR^{-/-} controls (~4-5 Log₁₀ at 48 hpi) highlighting the importance of interferon- α/β signaling in the control of ONNV infection as observed in other alphavirus animal models [42, 43, 51, 52]. Nonetheless, type I interferon induced upon *Plasmodium* infection seems to be negligible for the establishment of protective effects by murine malaria

as co-infected IFN α R $^{-/-}$ mice still displayed reduced ONNV infection to a comparable level than co-infected wild type mice (Fig S6).”

Minor issues:

Original point 2: The authors should consider including the supplementary data presented in the rebuttal regarding effects on malarial populations in the Supplementary Results.

Response: We thank the reviewer for the comment. We have included a new paragraph (lines 141-151) in the results section and a new supplementary figure (Figure S2) describing the effects of co-infection on malarial growth and survival.

“The effects of ONNV infection on the dynamics of malarial growth and survival were also assessed. The inoculation of ONNV four days post Py17x or PbA infection did not alter parasitemia levels (Figure S2A-B) or PbA-induced mortality (Figure S2B). Concurrent co-infection with non-lethal Py17x and ONNV resulted in increased parasitemia levels (Figure S2C). Py17x parasitemia resolution was delayed as co-infected animals took 26-28 days to clear blood-stage parasites compared to 20-22 days in controls. On the other hand, simultaneous inoculation with PbA and ONNV did not affect the development of parasitemia or ECM mortality in co-infected mice (Figure S2D). Finally, the infection with Plasmodium parasites four days post ONNV inoculation resulted in aggravated Py17x and PbA parasitemia (Figure S2E-F) but did not impact ECM mortality (Figure S2F)”.

Original point 3: The authors' response regarding distinctions in immune responses to different viruses (and supporting reference) should be incorporated in the Discussion.

Response: We have incorporated our response regarding distinctions in immune responses to different viruses in the discussion section (lines 384-395).

“A recent study [58] showed that Plasmodium infection protected mice from Ebola virus (EBOV)-induced mortality via upregulation of IFN γ , supporting field reports where co-infected patients by EBOV and *P. falciparum* displayed increased survival rates [59]. Conversely, two other murine co-infection models with respiratory viral pathogens such as murine pneumonia virus (PVM) and murine gammaherpesvirus 68 (MHV68) using non-lethal *P. chabaudi* and *P. yoelii* 17XNL have reported detrimental outcomes for the host such as increased viral loads in the lungs [60] and mortality due to severe anemia [61]. These observations were linked to altered type I interferon production [60] and antiviral humoral responses [61] upon co-infection. Thus, the protective or detrimental effects of murine malaria on viral pathogens is likely associated to the modulation of distinct immune responses governing the control of different viral infections”.

Original point 6: The authors' response should be incorporated into the Discussion. There are many dynamics to this system that are not obvious, and the apparent disconnect between circulating and tissue cytokine levels and how the two relate to immune protection is one of them.

Response: We have incorporated our response regarding differences in cytokines concentrations in footpad tissues and serum in the discussion section (lines 353-361).

“These observations strongly suggested that the timing of parasite inoculation and induction of IFN γ are critical for the protective effects to happen. Interestingly, although the main suppression of ONNV infection occurred in joint footpad cells, we observed lower concentrations of IFN γ in joint footpad tissues compared to serum samples at 4 days post-Plasmodium inoculation. Thus, it is likely that IFN γ levels in joint footpads could have increased at an earlier time point. In support of this, we observed high concentrations of IFN γ -induced immune mediators in joint footpad tissues, particularly CXCL10 [53] and CCL7, known to be produced by fibroblasts and mononuclear cells upon IFN γ stimulation [54]”.

January 4, 2022

RE: Life Science Alliance Manuscript #LSA-2021-01272-TRR

Prof. Lisa F.P. Ng
A*STAR Infectious Diseases Labs
Laboratory of Microbial Immunity
8A Biomedical Grove
#04-06 Immunus Building
Singapore 138648

Dear Dr. Ng,

Thank you for submitting your Research Article entitled "Malaria abrogates O'nyong-nyong virus pathologies by restricting virus infection in non-immune cells". It is a pleasure to let you know that your manuscript is now accepted for publication in Life Science Alliance. Congratulations on this interesting work.

DISTRIBUTION OF MATERIALS:

Again, congratulations on a very nice paper. I hope you found the review process to be constructive and are pleased with how the manuscript was handled editorially. We look forward to future exciting submissions from your lab.

Sincerely,
